# Smith–Magenis syndrome protein RAI1 regulates body weight homeostasis through hypothalamic BDNF-producing neurons and neurotrophin downstream signalling

Sehrish Javed[1,2], Ya-Ting Chang[1,2], Yoobin Cho[1,2], Yu-Ju Lee[1,2], Hao-Cheng Chang[1,2], Minza Haque[1,2], Yu Cheng Lin[1,2], Wei-Hsiang Huang[1,2]*

[1]Department of Neurology and Neurosurgery, Centre for Research in Neuroscience, McGill University, Montréal, Canada; [2]Brain Repair and Integrative Neuroscience Program, The Research Institute of the McGill University Health Centre, Montréal, Canada

**Abstract** *Retinoic acid-induced 1* (*RAI1*) haploinsufficiency causes Smith–Magenis syndrome (SMS), a genetic disorder with symptoms including hyperphagia, hyperlipidemia, severe obesity, and autism phenotypes. RAI1 is a transcriptional regulator with a pan-neural expression pattern and hundreds of downstream targets. The mechanisms linking neural *Rai1* to body weight regulation remain unclear. Here we find that hypothalamic brain-derived neurotrophic factor (BDNF) and its downstream signalling are disrupted in SMS (*Rai1*[+/-]) mice. Selective *Rai1* loss from all BDNF-producing cells or from BDNF-producing neurons in the paraventricular nucleus of the hypothalamus (PVH) induced obesity in mice. Electrophysiological recordings revealed that *Rai1* ablation decreased the intrinsic excitability of PVH[BDNF] neurons. Chronic treatment of SMS mice with LM22A-4 engages neurotrophin downstream signalling and delayed obesity onset. This treatment also partially rescued disrupted lipid profiles, insulin intolerance, and stereotypical repetitive behaviour in SMS mice. These data argue that RAI1 regulates body weight and metabolic function through hypothalamic BDNF-producing neurons and that targeting neurotrophin downstream signalling might improve associated SMS phenotypes.

*For correspondence:
wei-hsiang.huang@mcgill.ca

Competing interest: The authors declare that no competing interests exist.

## eLife assessment

This **valuable** study informs whether diminishing BDNF expression or alterations in the activity of BDNF-containing neurons in the paraventricular nucleus of the hypothalamus contributes to metabolic alterations in individuals with reduced RAI1 function, including those afflicted with Smith–Magenis syndrome (SMS). The evidence supporting the conclusions is **compelling** in that RAI1 deficits in BDNF-containing neurons partly contribute, with prominent effects on glycaemic control and modest effects on feeding and body weight regulation. This study would be of interest to neuroscientists and medical biologists working on metabolic disorders such as obesity and diabetes, as the findings in this study further link SMS-associated obesity with reduced Bdnf gene expression in the PVH and shed light on the role of the Rai1 gene in the PVH Bdnf neurons and offer a basis for future therapeutic strategies for managing obesity in SMS.

## Introduction

Obesity is an increasing global concern: 20% of the world's adult population is expected to become obese by 2025 if current trends persist (*Collaboration NCDRF, 2017*). However, the complex environmental, behavioural, and genetic contributions to obesity make this a complex condition to model and address. Monogenic obesity disorders provide a consistent system where the pathophysiology, genetic predisposition, and neurobiology underlying phenotypes can potentially provide insights that translate to polygenic obesity (*Loos and Yeo, 2022*).

One such monogenic disorder is Smith–Magenis syndrome (SMS; OMIM #182290) (*Smith et al., 1986*), a neurogenetic condition frequently associated with obesity. Eighty percent of SMS cases are caused by interstitial deletions of the 17p11.2 chromosomal region containing a dosage-sensitive gene, *retinoic acid-induced 1* (*RAI1*) (*Slager et al., 2003*). Most SMS patients are overweight (40% >95th percentile weight, 80% >85th percentile weight) and exhibit overeating tendencies by adolescence (*Alaimo et al., 2015*; *Burns et al., 2010*; *Gandhi et al., 2022*). Twenty percent of SMS patients do not have 17p11.2 deletions but instead carry *RAI1* point mutations that are still frequently associated with obesity (*Boot et al., 2021*; *Edelman et al., 2007*). These clinical data indicate that RAI is critical for body weight regulation.

The human RAI1 protein shares >80% sequence identity with the mouse RAI1 protein, allowing us to study RAI1 function in vivo (*Huang et al., 2016*; *Javed et al., 2020*). We previously found that *Rai1* encodes a transcriptional regulator that binds to the promoter or enhancer regions of its target genes, most of which are involved in neurodevelopment and synaptic transmission (*Huang et al., 2016*). SMS (*Rai1$^{+/-}$*) mice exhibit obesity associated with hyperphagia and increased adiposity (*Burns et al., 2010*; *Huang et al., 2016*). Interestingly, RAI1 may have a sexually dimorphic function as female SMS mice exhibit more pronounced obesity than males (*Burns et al., 2010*; *Huang et al., 2016*).

Using a homozygous *Rai1* conditional knockout (*Rai1*-cKO) mice, we found that *Rai1* deletion in either subcortical excitatory neurons (vGlut2$^+$) or single-minded homolog 1 (Sim1$^+$) neurons induced obesity in mice (*Huang et al., 2016*). Sim1$^+$ neurons reside in several brain regions including the nucleus of the lateral olfactory tract, the supraoptic nucleus, the medial amygdala, and the paraventricular nucleus of the hypothalamus (PVH), a region that is critical for body weight regulation (*Balthasar et al., 2005*). Sim1$^+$ neurons in the PVH are composed of several molecularly defined cell types, including those that express oxytocin, vasopressin, corticotropin-releasing hormone, somatostatin, melanocortin-4-receptor (MC4R), growth hormone-releasing hormone (GHRH), or brain-derived neurotrophic factor (BDNF) (*An et al., 2015*). Deleting *Bdnf* from the PVH results in obesity due to increased food intake, reduced energy expenditure, and reduced locomotion (*An et al., 2015*). While PVH neuronal activity is reduced during feeding (*Xu et al., 2019*), activation of PVH$^{BDNF}$ neurons results in weight loss (*Wu and Xu, 2022*). We recently found that RAI1 promotes *Bdnf* expression in the hypothalamus throughout life (*Javed et al., 2021*). We hypothesized that RAI11 may be connected to BDNF-producing neurons in PVH, explaining how a neural transcription factor regulates body weight homeostasis.

BDNF regulates energy metabolism by binding to its cognate receptor tropomyosin receptor kinase B (TRKB) (*An et al., 2015*) and may thus provide a potential therapeutic target for SMS. However, the link between *RAI1, BDNF*, and SMS symptoms is unclear. Patients carrying monogenic mutations in either *BDNF* (*Gray et al., 2006*) or *NTRK2* (encoding TRKB) (*Yeo et al., 2004*) experience severe hyperphagic obesity. Moreover, obesity and associated metabolic dysfunctions such as impaired glucose and insulin sensitivity are correlated with impaired PI3K-AKT pathway, a TRKB downstream target (*Su et al., 2021*). Given that (1) *Bdnf* overexpression during early embryonic development or (2) targeting *Bdnf* overexpression to the PVH of adolescent SMS (*Rai1$^{+/-}$*) mice protected them from weight gain and metabolic defects (*Javed et al., 2021*), and that (3) a point mutation in *RAI1* disrupts its ability to transcriptionally enhance *BDNF* expression in vitro (*Abad et al., 2018*), pharmacological targeting of neurotrophin downstream signalling is an attractive and yet unexplored therapeutic avenue for SMS.

Building on our previous discoveries, here we examine the neurobiological functions of RAI1 in BDNF-producing neurons, specifically those that reside in the PVH. Importantly, we evaluated the involvement of the BDNF-TRKB pathways in SMS pathogenesis using a small molecule drug (LM22A-4) to enhance neurotrophin downstream signalling. Our work indicates that PVH$^{BDNF}$ neurons depend on

RAI1 to maintain normal neuronal activity and regulate body weight homeostasis, and dysfunction of BDNF downstream signalling contributes to obesity associated with SMS mice.

## Results

### *Rai1* deficiency dysregulates specific downstream targets of neurotrophin in the mouse hypothalamus

Multiple lines of evidence indicated that SMS mice show decreased *Bdnf* mRNA expression in the hypothalamus (*Burns et al., 2010*; *Huang et al., 2016*; *Javed et al., 2021*). To determine if BDNF protein levels were consequently altered in the hypothalamus, we performed an enzyme-linked immunosorbent assay (ELISA) and found that BDNF protein levels were significantly reduced in 7-week-old pre-symptomatic SMS mice relative to controls (*Figure 1A*). To quantify if molecular pathways associated with BDNF downstream signalling were altered, we subjected hypothalamic tissue lysates from 7-week-old pre-obese SMS and control (Ctrl) mice to reverse-phase protein array (RPPA) (*Figure 1B*). Control and SMS samples are clustered separately in the principal component analysis (*Figure 1C*).

Hierarchical clustering analysis indicated that most differentially expressed phosphorylated proteins (*Figure 1D*) and total proteins (*Figure 1—figure supplement 1*) were downregulated in SMS mice. We mapped total proteins, as well as phospho-proteins, that were differentially expressed in Ctrl and SMS samples onto the interconnected biological network and identified a subset of connected nodes downregulated in SMS (FDR < 10e-5) belong to the neurotrophin, PI3K-AKT, and mTOR signalling cascades (*Figure 1E*). These molecules are known to play a critical role in regulating body weight homeostasis and can be activated by BDNF either directly (neurotrophin pathway) or indirectly (PI3K-AKT and mTOR pathways) (*Cota et al., 2006*; *Schultze et al., 2012*). The expression levels of PIK3CA, a common node in these pathways, were significantly reduced in the SMS mice compared to controls (*Figure 1F*). Moreover, the expression levels of phosphorylated NF-κB (nuclear factor-kappa B), a downstream target of the neurotrophin pathway, were significantly reduced in the SMS group (*Figure 1G*). Together, these results suggest that *Rai1* loss impairs specific pathways, including neurotrophin signalling, and indicate that dysfunction of BDNF-producing cells could contribute to SMS pathogenesis.

### BDNF-producing cells depend on RAI1 to regulate body weight homeostasis

To target BDNF-producing cells, we used a *Bdnf*^Cre/+ mouse line that carries a viral 2A oligopeptide and Cre recombinase immediately upstream of *Bdnf*'s endogenous STOP codon (*Tan et al., 2016*). We crossed the *Bdnf*^Cre/+ mice with tdTomato-Ai9 reporter mice and found that Ai9 signals faithfully reflected endogenous BDNF protein expression in the PVH (*Figure 2—figure supplement 1A–E*). We also found that RAI1 is expressed in PVH^BDNF cells that are distinct from larger magnocellular cells, such as the PVH oxytocin neurons (*Figure 2—figure supplement 2A–I*). Specifically, 55% of PVH^BDNF neurons express RAI1, whereas only 2% of PVH^Oxytocin neurons express RAI1 (*Figure 2—figure supplement 2I*). These data suggest that RAI1 is enriched in BDNF-producing neurons in the mouse PVH.

To determine if BDNF-producing cells depend on RAI1 to regulate body weight, we deleted one or both copies of *Rai1* using the *Bdnf*^Cre/+ and the conditional *Rai1*^fl/fl mice (*Huang et al., 2016*; *Figure 2A*, *Figure 2—figure supplement 3A*). Conditional knockout mice (cKO: *Bdnf*^Cre/+; *Rai1*^fl/fl) showed a significant reduction in Rai1 expression in PVH^BDNF neurons compared to controls (*Figure 2B and C*, *Figure 2—figure supplement 3B*). The total number of BDNF-producing cells in the PVH remained similar in Ctrl and cKO mice (*Figure 2—figure supplement 3C*), indicating that RAI1 loss did not affect PVH neurogenesis or survival.

We monitored body weight weekly and found that male and female cKO mice showed a significant increase in body weight beginning at 15 wk of age compared to controls (Ctrl: *Bdnf*^Cre/+) (*Figure 2D*, *Figure 2—figure supplement 4A*). However, conditional *Rai1* heterozygous mice (*Bdnf*^Cre/+; *Rai1*^fl/+) did not show a pathological increase in body weight (*Figure 2—figure supplement 3D*, *Figure 2—figure supplement 4B*), suggesting that non-BDNF-producing cells also contribute to SMS pathogenesis. EcoMRI found that 26-week-old cKO male and female mice showed increased fat mass but not lean mass (*Figure 2E*, *Figure 2—figure supplement 4C*). Further dissection of fat tissues from different organs showed an increased mass of subcutaneous inguinal (S.C. Ing.) fat in cKO mice of

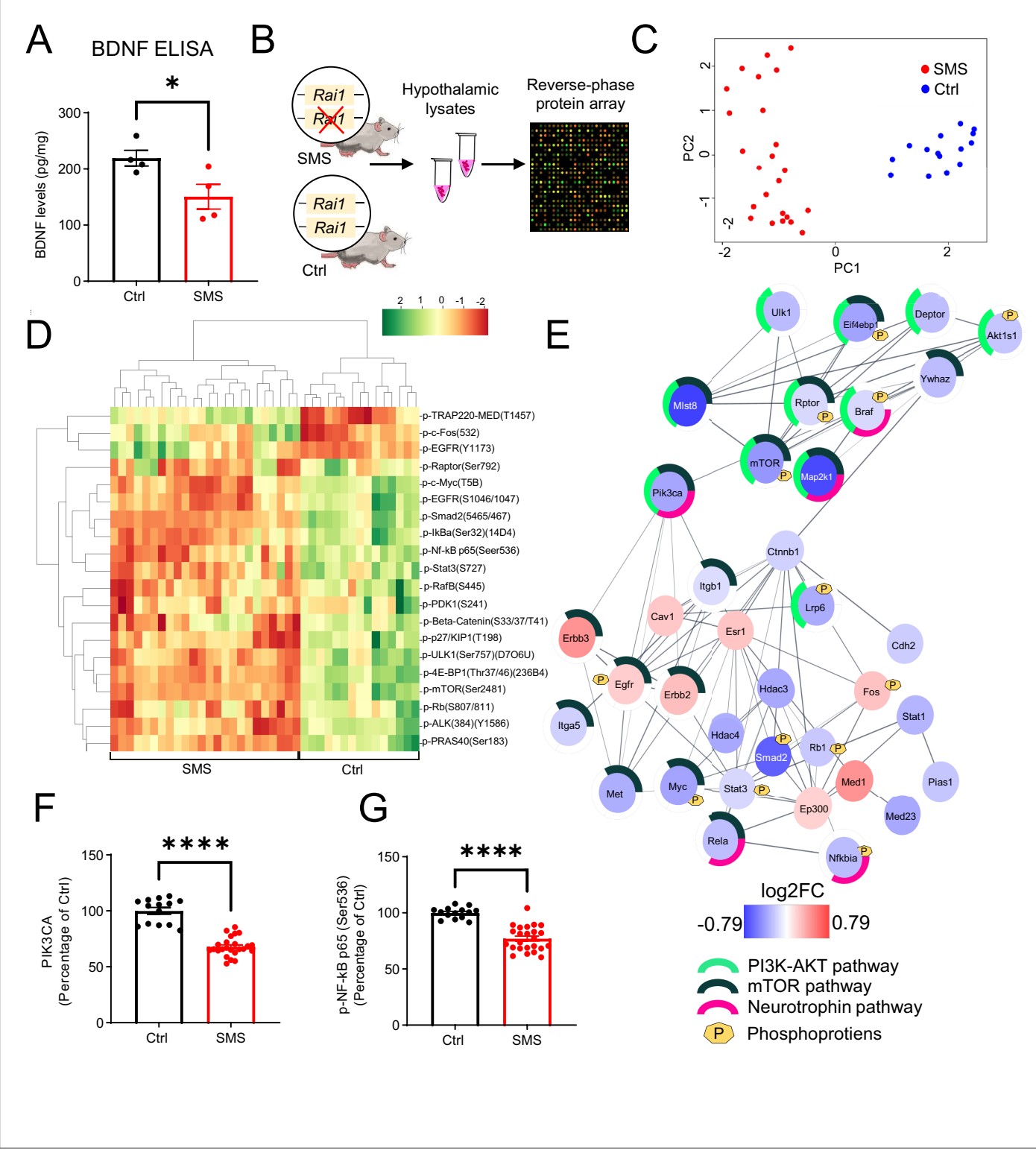

**Figure 1.** *Rai1* haploinsufficiency disrupts hypothalamic proteomic profile in mice. (**A**) ELISA showing that brain-derived neurotrophic factor (BDNF) protein levels in hypothalamic tissues were significantly reduced in Smith–Magenis syndrome (SMS) mice compared to controls (n = 4/genotype). *p<0.05, unpaired *t*-test, two-tailed. (**B**) A schematic diagram showing the experimental strategy. Hypothalamic tissue lysates from 7-week-old control (n = 5, technical triplicates/sample) and SMS (n = 8, technical triplicates/sample) mice were subjected to reverse-phase protein array (RPPA) that utilizes a cocktail of 240 validated antibodies to probe several key intracellular signalling pathways by quantifying the expression levels of total and phosphorylated proteins. (**C**) Principal component analysis segregates the SMS and Ctrl groups where PC1 explains 47% of the variance, and PC2

*Figure 1 continued on next page*

*Figure 1 continued*

explains 25% of the variance. (**D**) Heat map showing hierarchical clustering of differentially expressed phospho-proteins: most showed reduced levels in SMS samples. (**E**) Protein–protein interaction (PPI) analysis showing differentially expressed proteins and phospho-proteins in a crosstalk network. PPI enrichment p-value:<1.0e-16. The nodes represent the proteins/phospho-proteins and the lines indicate physical interactions. Associations with specific molecular pathways are shown as colour-coded outer rings. (**F, G**) The protein levels of two neurotrophin pathway-related nodes, PIK3CA (**F**) and phospho-NF-kB (**G**), were significantly reduced in SMS samples. ****p<0.0001, unpaired *t*-test, two-tailed.

The online version of this article includes the following figure supplement(s) for figure 1:

**Figure supplement 1.** Differentially expressed proteins in the hypothalamus of Smith–Magenis syndrome (SMS) mice.

both sexes and an increased mass of epididymal adipose tissues (eWAT) in female cKOs (*Figure 2F*, *Figure 2—figure supplement 4D*).

In contrast, the total mass of brown adipose tissue (BAT) remained unchanged between groups (*Figure 2F*, *Figure 2—figure supplement 4D*). In both male and female cKO mice, we found hypertrophic eWAT adipocytes (*Figure 2G*, *Figure 2—figure supplement 4E*) and elevated blood leptin levels (*Figure 2H*, *Figure 2—figure supplement 4F*). However, there was no significant alteration in blood lipids such as triglycerides (TG), high-density lipoprotein (HDL), low-density lipoproteins (LDL), and very low-density lipoprotein (VLDL) in either male or female cKO mice (*Figure 2—figure supplement 3E–G*, *Figure 2—figure supplement 4G–I*). These data suggest that obesity caused by ablating *Rai1* from BDNF-producing cells is associated with increased fat mass, fat deposition, and leptin levels.

Obesity could result from increased food intake, reduced energy expenditure, or both. Similar to SMS mice, male and female cKO mice did not have defects in respiratory exchange rates when measured by indirect calorimetry (*Figure 2—figure supplement 3H*, *Figure 2—figure supplement 4J*). Interestingly, cKO mice showed sex-specific differences in energy expenditure. During the dark phase, female cKO mice showed reduced energy expenditure (*Figure 2I*) and decreased total locomotor activity (*Figure 2J*). In contrast, total food intake remained unchanged (*Figure 2K*). Male cKO mice became hyperphagic (*Figure 2—figure supplement 4K*), showed increased energy expenditure during the dark phase (*Figure 2—figure supplement 4L*), and exhibited normal locomotor activity when compared to controls (*Figure 2—figure supplement 4M*). These results suggest that obesity in male cKO mice is attributable to overeating despite increased energy expenditure.

To determine if *Rai1* loss from BDNF-producing cells impacts metabolic function, a glucose tolerance test (GTT) was performed. Interestingly, cKO mice of both sexes showed normal blood glucose levels right before the test but became hyperglycaemic 20 min (female cKO) and 45 min (male cKO) after glucose administration (*Figure 2L*, *Figure 2—figure supplement 4N*). This change was accompanied by a significantly higher area under the curve (AUC) for both groups (*Figure 2—figure supplement 3I*, *Figure 2—figure supplement 4O*). While the male cKO mice remained hyperglycaemic throughout the GTT (*Figure 2—figure supplement 4N*), the glucose levels in female cKO mice returned to normal towards the end (*Figure 2L*). We also found increased blood insulin levels during the GTT in both sexes (*Figure 2—figure supplement 3J*, *Figure 2—figure supplement 4P*). In the insulin tolerance test (ITT), female cKO mice showed moderately elevated glucose levels 20 min into the test (*Figure 2—figure supplement 3K*). By contrast, male cKO mice showed considerably higher glucose levels than controls in the ITT (~6.5 ng/ml for cKO and ~2.0 ng/ml for Ctrl) during the 60–120 min intervals (*Figure 2—figure supplement 4Q*). These experiments demonstrate that BDNF-producing cells depend on RAI1 to regulate body weight, adiposity, glucose, and insulin tolerance in mice.

### *Rai1* deletion reduces PVH^BDNF neuronal excitability

BDNF-producing neurons are widely distributed in the hypothalamus, being found in the dorsomedial hypothalamus (*Unger et al., 2007*), ventromedial hypothalamus (*Xu et al., 2003*), and PVH (*An et al., 2015*). To understand how *Rai1* loss affects intrinsic neuronal excitability and synaptic transmission of PVH^BDNF neurons that underlie energy homeostasis (*An et al., 2015*), we fluorescently labelled them with Cre-dependent tdTomato (*Figure 3A*). We found that a hyperpolarized step followed by increasing the holding voltage at resting membrane potentials (Rm) was able to initiate rebound firing or rebound repetitive firing in most PVH^BDNF neurons (*Figure 3—figure supplement 1A*). Hyperpolarization-induced $I_H$ currents and low depolarization events were observed in some PVH^BDNF neurons (*Figure 3—figure supplement 1A*). cKO PVH^BDNF neurons showed reduced action potential

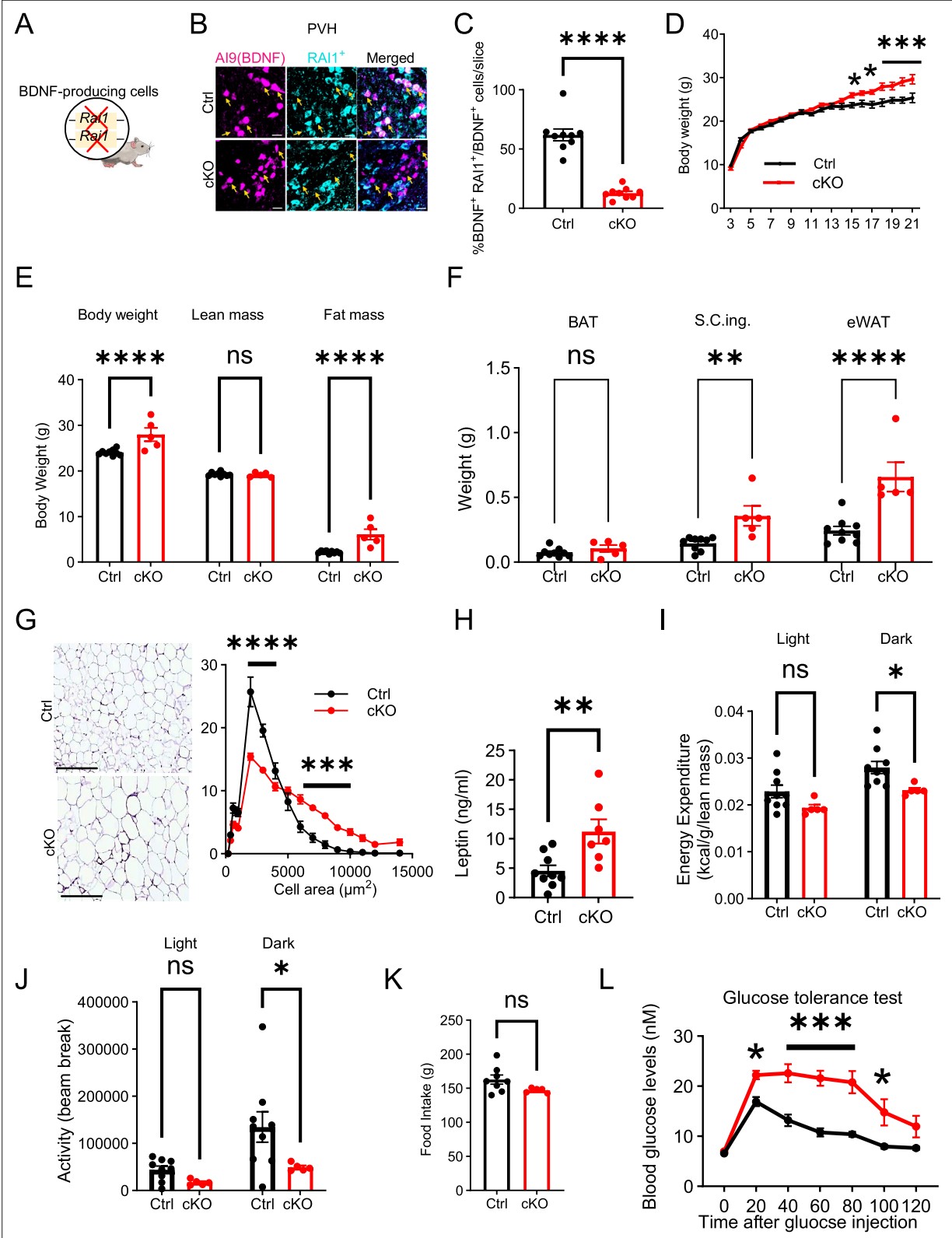

**Figure 2.** RAI1 is required in brain-derived neurotrophic factor (BDNF)-producing cells to regulate energy metabolism. (**A**) A schematic diagram showing selective deletion of *Rai1* from BDNF-producing cells in mice. (**B**) Representative images showing that in Ctrl mice (*Bdnf$^{Cre/+}$; Ai9*), many PVH$^{BDNF}$ neurons (magenta) express RAI1 (cyan, double-positive cells are indicated with yellow arrows). By contrast, PVH$^{BDNF}$ neurons lack RAI1 expression in the conditional knockout (cKO) group (*Bdnf$^{Cre/+}$; Rai1$^{fl/fl}$; Ai9*). Scale bars = 20 µm. (**C**) Percentage of PVH$^{BDNF}$ neurons co-expressing RAI1 in Ctrl (n = 3) and

*Figure 2 continued on next page*

*Figure 2 continued*

cKO (n = 3) mice. unpaired *t*-test, two-tailed, ****p<0.0001. (**D**) Female cKO mice (n = 12) showed a significant weight gain when compared to female Ctrl mice (n = 10). Two-way ANOVA with Šidák's multiple comparisons test. *p<0.05, ***p<0.001. (**E**) Body composition was measured with echo-MRI, showing an increased fat mass in 26-week-old mice. ns indicates the difference is not significant. Two-way ANOVA with Šidák's multiple comparisons test. ****p<0.0001. (**F**) Fat mass of brown adipocytes (BAT), subcutaneous inguinal (S.C.ing) and epididymal white adipose tissue (eWAT) in Ctrl and cKO mice. ns indicates the difference is not significant. Two-way ANOVA with Šidák's multiple comparisons test. **p<0.01, ****p<0.0001. (**G**) Representative images showing eWAT adipocyte hypertrophy of the cKO mice (left). Scale bar = 500 μm. Frequency distribution of adipocytes at each cellular size (right) (Ctrl: n = 4, cKO: n = 5). Two-way ANOVA with Šidák's multiple comparisons test. ***p<0.001, ****p<0.0001. (**H**) Female cKO mice showed significantly increased blood leptin levels. unpaired *t*-test, two-tailed, **p<0.01. (**I**) Female cKO mice showed reduced energy expenditure during the dark phase. ns indicates not significantly different. Two-way ANOVA with Šidák's multiple comparisons test. *p<0.05. (**J**) Female cKO mice showed reduced locomotor activity during the dark phase. ns indicates the difference is not significant. Two-way ANOVA with Šidák's multiple comparisons test. *p<0.05. (**K**) Female cKO mice showed similar food intake as control mice. ns indicates not significant. unpaired *t*-test, two-tailed. (**L**) Glucose tolerance test showing that female cKO mice became hyperglycaemic 20 min after intraperitoneal glucose administration, suggesting glucose intolerance (Ctrl: n = 9, cKO: n = 7). Two-way ANOVA with Šidák's multiple comparisons test. *p<0.05, ***p<0.001. Data are shown as mean ± SEM.

The online version of this article includes the following figure supplement(s) for figure 2:

**Figure supplement 1.** The *Bdnf^Cre* allele labels brain-derived neurotrophic factor (BDNF)-producing neurons in the paraventricular nucleus of the hypothalamus.

**Figure supplement 2.** RAI1 expression in brain-derived neurotrophic factor (BDNF)-expressing but not oxytocin-expressing magnocellular paraventricular nucleus of the hypothalamus (PVH) neurons.

**Figure supplement 3.** Ablating *Rai1* from the brain-derived neurotrophic factor (BDNF)-producing cells induces body weight gain and defective energy homeostasis.

**Figure supplement 4.** Metabolic profiles of male conditional knockout (cKO) mice lacking RAI1 expression in the brain-derived neurotrophic factor (BDNF)-producing cells.

(AP) firing frequency at a holding voltage of –60 mV compared to Ctrl PVH^BDNF neurons (***Figure 3B and C***). The resting membrane potential (Rm) of cKO (–52 mV) neurons showed a 10 mV positive shift compared to Ctrl neurons (–62 mV) (***Figure 3D***). We also found that fewer cKO PVH^BDNF neurons showed spontaneous firing at Rm (***Figure 3E***, Ctrl:10/13; cKO:5/12) and at a holding voltage of –70 mV (***Figure 3F***, Ctrl:6/12; cKO:1/12) compared to Ctrl PVH^BDNF neurons. The threshold of the first AP initiated by a ramp current injection (1 nA/s) from Rm was significantly increased in cKO neurons (***Figure 3G and H***). Because a wider shoulder and lower AP amplitude were observed when held at more depolarized potentials (***Figure 3—figure supplement 1B***), we compared the AP waveforms at a holding voltage of –60 mV, close to the Rm of both genotypes. The AP of the cKO group exhibited a lower peak amplitude (measuring from the threshold to peak) (***Figure 3I***), a normal threshold (***Figure 3J***), and a normal half-width (***Figure 3—figure supplement 1C***). In summary, we found that both passive and AP properties of cKO PVH^BDNF neurons were altered due to RAI1 loss. The Rm of cKO neurons shifted to a more positive voltage than the gating threshold of voltage-gated sodium channels (6/12 were more positive than –55 mV), which was unfavourable for their transition from the inactivation state to the closed state. Combined with the higher AP threshold of cKO neurons, these changes contribute to a decreased excitability in cKO PVH^BDNF neurons.

PVN neurons receive glutamatergic and GABAergic inputs and are tonically inhibited by GABAergic neurons (***Colmers and Bains, 2018***; ***Gao et al., 2017***). We observed a slight rightward shift of the probability of miniature inhibitory postsynaptic current (mIPSC) frequency in cKO PVH^BDNF neurons, although the average frequency (***Figure 3K***) was not significantly different between groups. The probability of mIPSC amplitude also showed a right shift without a significant change (***Figure 3L***, ***Figure 3—figure supplement 1D***). However, we observed a significantly increased AUC (***Figure 3M***). Interestingly, we found that cKO PVH^BDNF neurons had a smaller cellular diameter (***Figure 3—figure supplement 1E***). Overall, our data argue that *Rai1* deletion decreases the excitability and AP firing of PVH^BDNF neurons without significant changes in synaptic inhibitory transmission.

## RAI1 is necessary for PVH^BDNF neurons to regulate energy homeostasis

Reduced PVH^BDNF neuronal activity in cKO mice suggested that RAI1 loss in BDNF-producing PVH neurons might be sufficient to induce defective energy homeostasis in vivo. To test this hypothesis by deleting *Rai1* only in PVH^BDNF neurons but not other BDNF-producing cells, we performed PVH^BDNF-specific *Rai1* deletion using a CRISPR/Cas9 system packaged into AAV9. Two viruses were delivered:

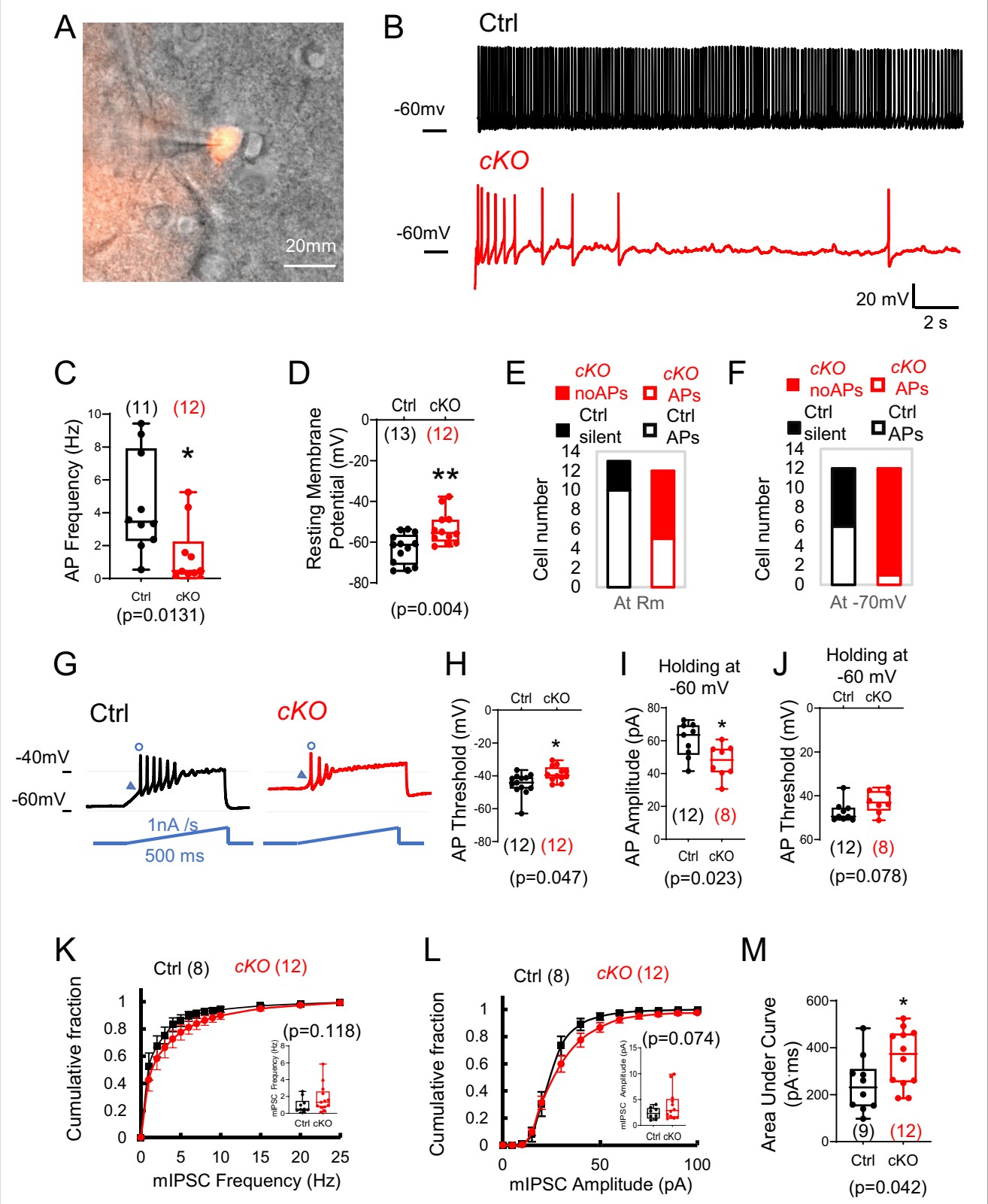

**Figure 3.** *Rai1* loss reduces intrinsic neuronal excitability and enhances inhibitory synaptic transmission of PVH[BDNF] neurons. (**A**) A representative image showing a patched PVH[BDNF] neuron labelled by *Bdnf[Cre]*-dependent tdTomato fluorescence signals. (**B, C**) Representative traces of spontaneous firing of control (black) and conditional knockout (cKO) (red) neurons at a holding voltage of –60 mV (**B**). The average spontaneous action potential (AP) firing frequencies are shown in (**C**). *p<0.05, unpaired Student's *t*-tests. (**D**) The resting membrane potentials of control (black) and cKO (red) PVH[BDNF] neurons.

*Figure 3 continued on next page*

Figure 3 continued

**p<0.01, unpaired Student's *t*-tests. (**E, F**) The number of control (black) and cKO (red) PVH<sup>BDNF</sup> neurons that showed spontaneous firing (open) or are silent (solid) at resting membrane potentials (**E**) and a holding voltage of –70 mV (**F**). (**G**) Representative traces of elicited APs in control (black) and cKO (red) PVH<sup>BDNF</sup> neurons responding to an inject current (bottom, blue), which ramps up from the resting membrane potentials at 1 nA/s for 500 ms. (**H**) The threshold of the first AP initiated by a current ramp in control (black) and cKO (red) PVH<sup>BDNF</sup> neurons. *p<0.05, unpaired Student's *t*-tests. (**I, J**) The amplitude (**I**) and threshold (**J**) for the first AP firing at the holding voltage of –60 mV in control (black) and cKO (red) PVH<sup>BDNF</sup> neurons. *p<0.05, unpaired Student's *t*-tests. (**K**) The cumulative fractions of miniature inhibitory postsynaptic current (mIPSC) frequency were measured in control (black) and cKO (red) PVH<sup>BDNF</sup> neurons. The average frequency of events from a 2 min stable recording of each neuron is shown in the inset, unpaired Student's *t*-tests. (**L**) The cumulative fractions of mIPSC amplitude of control (black) and cKO (red) PVH<sup>BDNF</sup> neurons. The average amplitude of events from a 2 min stable recording of each neuron is normalized to its capacitance and shown in the inset, unpaired Student's *t*-tests. (**M**) The area under the curve of each mIPSC event from a 2 min recording is averaged. Controls are shown in black, and cKOs are shown in red. unpaired *t*-tests, two-tailed. *p<0.05. Data are shown as mean ± SEM. Statistics: neuron numbers are in brackets (more than three mice/genotype).

The online version of this article includes the following figure supplement(s) for figure 3:

**Figure supplement 1.** The cellular and synaptic properties of control and *Rai1*-deficient PVH<sup>BDNF</sup> neurons.

one encoding a reporter molecule (GFP) and three single-guide RNAs targeting the *Rai1* gene (sgRai1), and the other carrying Cre-dependent *Staphylococcus aureus* CRISPR-associated protein 9 (saCas9) (*Figure 4A*). Hereafter, our experiments focus on female SMS mice since they showed more consistent defects in fat distribution (*Burns et al., 2010*) and body weight regulation (*Huang et al., 2016*) than males. Bilateral administration of two adeno-associated viruses (AAVs) into the PVH of female *Bdnf*<sup>Cre/+</sup> mice (Exp group) at 3 wk of age resulted in a significant loss of Rai1 protein expression in PVH<sup>BDNF</sup> neurons when compared to *Bdnf*<sup>Cre/+</sup> mice injected with an AAV encoding the *sgRai1* and GFP (Ctrl group) (*Figure 4B and C*, *Figure 4—figure supplement 1A*). Quantification of AAV-infected (GFP<sup>+</sup>) BDNF neurons showed that while the total number of BDNF<sup>+</sup> (*Figure 4—figure supplement 1B*) and BDNF<sup>+</sup> GFP<sup>+</sup> (*Figure 4—figure supplement 1C*) cells were not different in the two groups, the percentage of AAV-infected BDNF<sup>+</sup> cells that express RAI1 was significantly reduced in the Exp group (*Figure 4C*, *Figure 4—figure supplement 1D and E*). Moreover, the virus expression did not spread to the non-PVH neighbouring nuclei (*Figure 4—figure supplement 1F*). Deleting *Rai1* from the PVH<sup>BDNF</sup> neurons resulted in a significant increase in body weight beginning at 17 wk of age (*Figure 4D*). Similar to the cKO (*Bdnf*<sup>Cre/+</sup>; *Rai1*<sup>fl/fl</sup>) mice, obesity in 26-week-old Exp mice was due to increased fat mass (*Figure 4E*) and more specifically, increased subcutaneous inguinal (S.C. ing) and epididymal white adipose tissues (eWAT) but not BAT (*Figure 4F*). We also found that eWAT adipocytes became hypertrophic (*Figure 4G*). Similar to the cKO mice, food intake did not differ between Exp and Ctrl groups (*Figure 4H*), likely due to the high level of satiety signals in the brain caused by increased blood leptin levels (*Figure 4I*). We also found that energy expenditure (*Figure 4J*), respiratory exchange rate (*Figure 4—figure supplement 1G*), and blood lipid parameters, including TG (*Figure 4—figure supplement 1H*), HDL (*Figure 4—figure supplement 1I*), LDL and VLDL levels (*Figure 4—figure supplement 1J*), were similar in Exp and Ctrl groups, consistent with the cKO data (*Figure 2*). Interestingly, Exp mice showed significantly less locomotor activity during the dark phase than the Ctrl mice (*Figure 4K*). A GTT indicated that Exp mice have a hyperglycaemic profile (*Figure 4L*), accompanied by a slightly higher AUC for blood glucose levels (p=0.05) (*Figure 4M*). Insulin levels in Exp mice were also significantly elevated 15 min after glucose administration (*Figure 4N*).

While Exp mice showed a trend towards increased glucose levels in ITTs, the change was not statistically different (*Figure 4—figure supplement 1K*, 5). Altogether, these experiments indicate that conditional deletion of *Rai1* from PVH<sup>BDNF</sup> neurons during adolescence impairs the regulation of body weight, adiposity, glucose tolerance, and locomotion. These changes recapitulate most phenotypes observed in cKO mice with *Rai1* deleted in BDNF-producing cells from early development.

## Pharmacologically enhancing neurotrophin downstream signalling partially alleviates obesity in SMS mice

Our findings so far indicate that multiple pathways, including neurotrophin signalling, are dysregulated in SMS mice, and that Rai1 loss from BDNF-producing cells, including PVH<sup>BDNF</sup> neurons, impairs energy homeostasis in vivo. Next we sought to examine the contribution of BDNF dysfunction in SMS pathogenesis by pharmacologically activating downstream targets of the neurotrophin pathway.

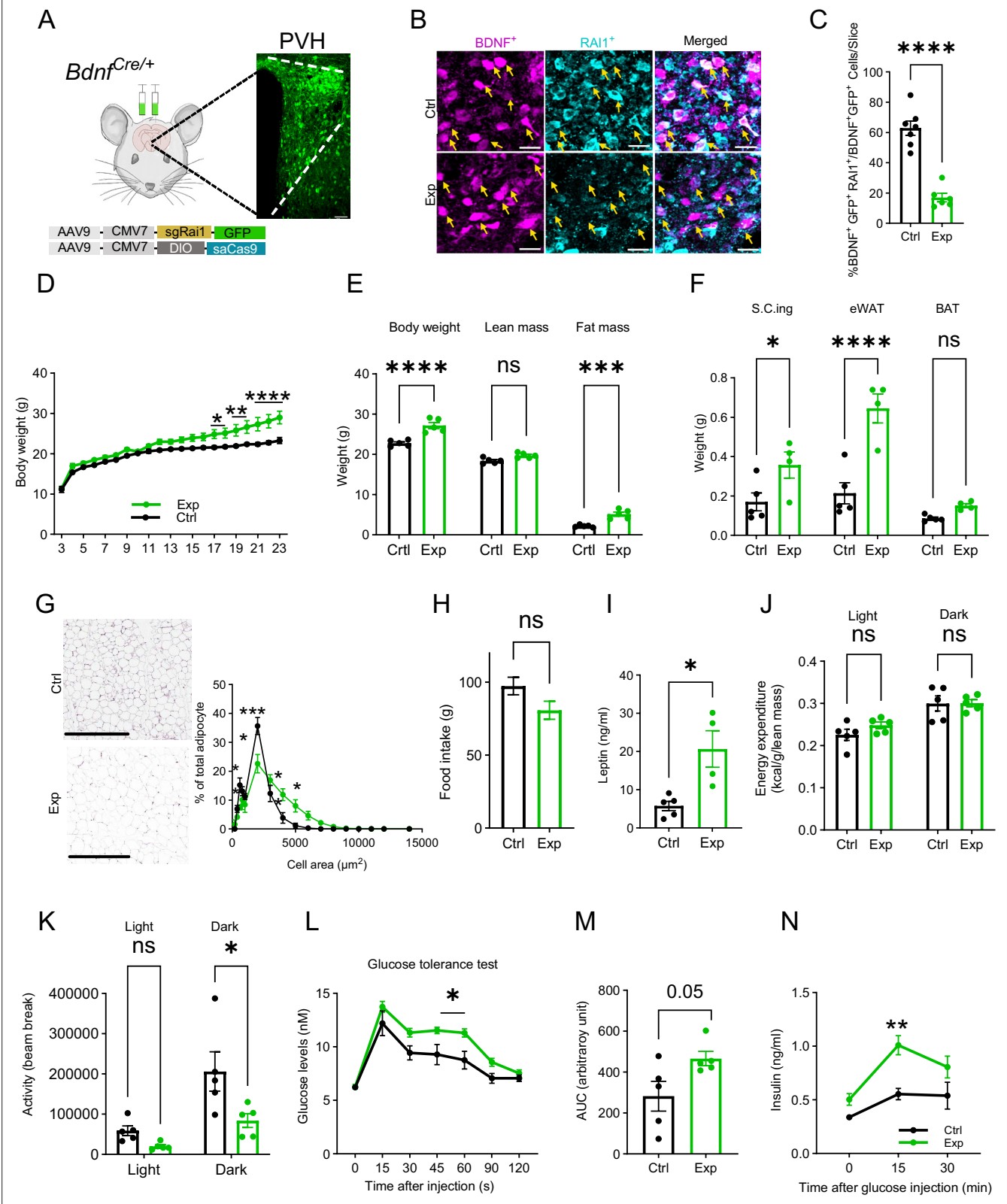

**Figure 4.** Selective *Rai1* ablation in PVH[BDNF] neurons induces obesity. (**A**) A schematic showing stereotaxic injection of adeno-associated viruses (AAVs) expressing Cas9 protein and single-guide RNAs targeting *Rai1* (sgRai1) to delete *Rai1* from the PVH[BDNF] neurons of *Bdnf*[Cre/+] female mice. Scale bar = 50 µm. (**B**) Representative images showing that RAI1 immunoreactivity (cyan) was dramatically reduced in *Bdnf*[Cre/+]; *Ai9* mice injected with both sgRai1 and DIO-saCas9 viruses (Exp group n = 3, bottom row) into the paraventricular nucleus of the hypothalamus (PVH). In contrast, many PVH[BDNF] neurons

*Figure 4 continued on next page*

*Figure 4 continued*

in *Bdnf*$^{Cre/+}$; *Ai9* mice injected only with the sgRai1 virus showed RAI1 expression (Ctrl group n = 3, top row). (**C**) The percentage of PVH$^{BDNF}$ neurons co-expressing RAI1 and GFP (virus) was significantly reduced in the Exp group, indicating a successful *Rai1* deletion. unpaired *t*-test, two-tailed, ****p<0.0001 (**D**) PVH$^{BDNF}$ neuron-specific *Rai1* deletion induced body weight gain in Exp mice (Ctrl group: n = 5, Exp group: n = 5). Two-way ANOVA with Šidák's multiple comparisons test. *p<0.05, **p < 0.01, ****p < 0.0001. (**E**) Body composition was measured with Echo-MRI, showing an increased fat mass deposition in 26-week-old Exp mice. ns indicates the difference is not significant. Two-way ANOVA with Šidák's multiple comparisons test. ***p<0.001, ****p < 0.0001. (**F**) Fat deposition analysis shows that the Exp group has significantly more subcutaneous inguinal (S.C.ing) and epididymal white adipose tissue (eWAT) mass (grams). ns indicates the difference is not significant. Two-way ANOVA with Šidák's multiple comparisons test. *p<0.05, ****p < 0.0001. (**G**) Representative images showing eWAT adipocyte hypertrophy of the Exp group (left). Scale bar = 500 μm. Frequency distribution of adipocytes at each cellular size (right). ns indicates not significantly different. Two-way ANOVA with Šidák's multiple comparisons test. *p<0.05, ***p<0.001. (**H**) Exp and control mice showed similar food intake. ns indicates not significantly different, unpaired *t*-test, two-tailed. (**I**) Blood leptin levels were significantly increased in the Exp mice. unpaired *t*-test, two-tailed, *p<0.05. (**J**) Exp and control mice showed similar energy expenditure. ns indicates the difference is not significant. Two-way ANOVA with Šidák's multiple comparisons test. (**K**) Exp mice showed reduced locomotor activity in the dark phase. ns indicates the difference is not significant. Two-way ANOVA with Šidák's multiple comparisons test. *p<0.05. (**L**) Exp mice became hyperglycaemic during the glucose tolerance test. ns indicates the difference is not significant. Two-way ANOVA with Šidák's multiple comparisons test. *p<0.05. (**M**) Measurement of the area under the curve (AUC) in Exp and Ctrl mice during the glucose tolerance test. unpaired *t*-test, two-tailed, p=0.05. (**N**) Exp mice showed increased plasma insulin levels during the glucose tolerance test. Two-way ANOVA with Šidák's multiple comparisons test, **p<0.01. Data are shown as mean ± SEM.

The online version of this article includes the following figure supplement(s) for figure 4:

**Figure supplement 1.** Metabolic profile of mice lacking *Rai1* in PVH$^{BDNF}$ neurons.

We systematically treated mice with LM22A-4, a potential small molecule BDNF loop domain mimetic (*Massa et al., 2010*). To measure the bioavailability of LM22A-4 in the SMS brain, we simultaneously administered LM22A-4 intranasally (5 mg/kg body weight) and intraperitoneally (50 mg/kg body weight) using the highest doses used in previous studies (*Canals et al., 2004*; *Chang et al., 2006*; *Gu et al., 2022*; *Kron et al., 2012*; *Ogier et al., 2007*). We then harvested forebrain and hypothalamic tissues 1 or 3 hr after drug administration. The spectrometric analysis found a significantly higher LM22A-4 concentration in both brain regions 1-hr post-treatment, while its concentration decreased 3 hr after treatment (*Figure 5—figure supplement 1A*). Consistent with the reduced neurotrophin signalling observed in our proteomic analysis, saline-treated SMS mice showed reduced phosphorylated-AKT (p-AKT; *Figure 5A*, *Figure 5—figure supplement 1B*, *Figure 5—source data 1*, *Figure 5—figure supplement 1—source data 1*) levels in the hypothalamus. LM22A-4 treatment significantly increased the levels of p-AKT in the SMS mice 1 hr post-treatment (*Figure 5A*, *Figure 5—figure supplement 1B*, *Figure 5—source data 1*, *Figure 5—figure supplement 1—source data 1*), indicating increased signalling downstream of the BDNF pathway.

We next examined if chronic LM22A-4 treatment was sufficient to reverse or modify the progression of the obesity phenotype in adult SMS mice. Specifically, we treated SMS mice daily from 8 to 14 wk of age, during which SMS mice began to gain significantly more weight. Saline-treated SMS mice (*Figure 5C*) were significantly heavier than saline-treated control mice from 9 wk of age onwards. Interestingly, LM22A-4-treated SMS mice showed a similar body weight to saline-treated control littermates until 13 wk of age but became obese by 14 wk of age (*Figure 5D*). Consistent with a partial rescue, LM22A-4-treated SMS mice were significantly less obese than saline-treated SMS mice throughout the treatment period (*Figure 5E*). Control mice treated with LM22A-4 did not show a further reduction in body weight (*Figure 5—figure supplement 1C*), indicating that the therapeutic effects were specific to SMS pathogenesis.

We measured food intake and found that LM22A-4 did not significantly rescue increased food intake in SMS mice (*Figure 5F*). SMS mice showed normal energy expenditure (*Figure 5—figure supplement 1D*) and respiratory exchange rate (*Figure 5—figure supplement 1E*), neither of which was altered by LM22A-4 treatment. Interestingly, we did observe a partial rescue of locomotor activity in LM22A-4-treated SMS mice that was not significantly different from saline-treated Ctrl or SMS mice (*Figure 5G*). We also found that the LM22A-4 treatment partially rescued fat deposition (*Figure 5H*) and completely rescued HDL levels (*Figure 5I*) in SMS mice. During an ITT, we found that saline-treated SMS mice showed increased baseline insulin levels (*Figure 5J*) and blood glucose levels during the test (*Figure 5K*), both of which were partially rescued by LM22A-4 treatment.

In addition to a deficit in energy homeostasis, SMS mice showed two highly reproducible behavioural deficits: increased stereotypic rearing in the open field and decreased social dominance

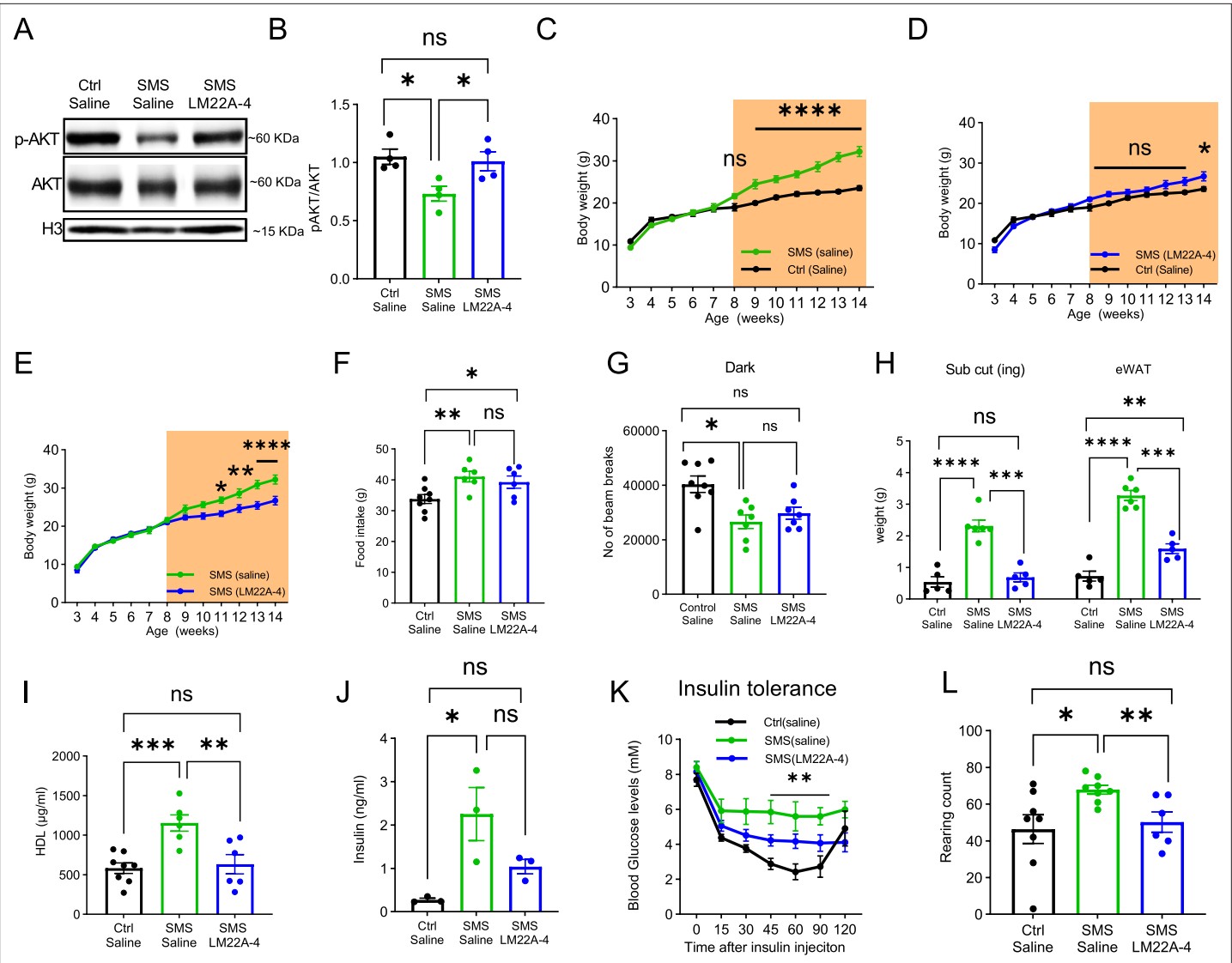

**Figure 5.** LM22A-4 treatment partially alleviates obesity and stereotypic behaviour in adult Smith–Magenis syndrome (SMS) mice. (**A**) Representative western blot showing p-AKT, total AKT, histone 3 (H3, a loading control) levels in the hypothalamus of saline-treated Ctrl, saline-treated SMS, and LM22A-4-treated SMS mice. (**B**) Quantification showing deficits in p-AKT levels of the hypothalamus were reversed by LM22A-4 treatment in SMS mice. Unpaired t-test, two-tailed, *p<0.05. (**C–E**) Body weight of saline-injected control (n = 8), saline-injected SMS (n = 8), and LM22A-4-injected SMS mice (n = 10). Saline or LM22A-4 treatment periods are highlighted in orange. ns indicates not significantly different. Two-way ANOVA with Šidák's multiple comparisons test. *p<0.05, **p<0.01, ****p<0.0001. (**F**) Food intake (in grams) of saline-injected control, saline-injected SMS, and LM22A-4-injected SMS mice. ns indicates the difference is not significant. unpaired t-test, two-tailed, *p<0.05, **p<0.01. (**G**) Locomotor activity (number of beam breaks) for saline-injected control, saline-injected SMS, and LM22A-4 injected SMS mice. ns indicates the difference is not significant. unpaired t-test, two-tailed, *p<0.05. (**H**) Fat deposition analysis shows that the saline-injected SMS group has significantly more subcutaneous inguinal (S.C.ing) and epididymal white adipose tissue (eWAT) mass (grams) than saline-injected Ctrl mice. S.C.ing and eWAT mass in LM22A-4-injected mice is comparable to the saline-injected mice. Two-way ANOVA with Šidák's multiple comparisons test. **p<0.01, ***p<0.001, ****p<0.0001. (**I**) High-density lipoprotein (HDL) levels were restored to normal in LM22A-4-injected SMS mice. ns indicates the difference is not significant. unpaired t-test, two-tailed, **p<0.01, ***p<0.001. (**J**) Insulin levels in saline-injected control, saline-injected SMS, and LM22A-4-injected SMS mice right before the start of the insulin tolerance test (time point 0). ns indicates the difference is not significant. unpaired t-test, two-tailed, *p<0.05. (**K**) Insulin tolerance test performed on saline-injected control (n = 8), saline-injected SMS ( n = 7), and LM22A-4-injected SMS mice (n = 7), 2 wk post daily administration of LM22A-4 or saline, shows reductions in the blood glucose levels of LM22A-4 treated SMS mice. * shows significance between saline-injected Ctrl and saline-injected SMS group. ns indicates the difference is not significant. Two-way ANOVA with Šidák's multiple comparisons test. **p<0.01. (**L**) Stereotypic rearing behaviour was fully rescued in LM22A-4-treated SMS mice. ns indicates the difference is not significant. unpaired t-test, two-tailed. *p<0.05, **p<0.01. Data are shown as mean ± SEM.

The online version of this article includes the following source data and figure supplement(s) for figure 5:

*Figure 5 continued on next page*

*Figure 5 continued*

**Source data 1.** Original uncropped western blot images for *Figure 5*.

**Figure supplement 1.** LM22A-4 treatment does not disrupt energy homeostasis and repetitive rearing in the Ctrl mice.

**Figure supplement 1—source data 1.** Original uncropped western blot images for *Figure 5—figure supplement 1*.

**Figure supplement 2.** LM22A-4 treatment in adult Smith–Magenis syndrome (SMS) mice is insufficient to improve social interaction deficit.

in the tube test (*Huang et al., 2018*; *Rao et al., 2017*). Saline-treated SMS mice showed significantly more rearing behaviour when compared to the saline-treated controls (*Figure 5L*). Interestingly, the LM22A-4 treatment reduced the repetitive rearing behaviour in SMS mice (*Figure 5L*). LM22A-4 treatment did not change repetitive behaviours in control mice (*Figure 5—figure supplement 1F*), suggesting the effect is specific to *Rai1* haploinsufficiency. However, we did not observe a rescue of social dominance deficits in SMS mice (*Figure 5—figure supplement 2A–E*). Together, these data indicate that LM22A-4 treatment delayed the onset of obesity and fully rescued repetitive behaviour in SMS mice, suggesting that neurotrophin downstream signalling dysfunction contributes to changes in the mouse SMS brain linked to obesity and behavioural phenotypes.

## Discussion

The hypothalamus and neurotrophin signalling in this region control body weight by regulating the intricate balance between energy intake and expenditure (*Xu and Xie, 2016*). The role of BDNF-producing neurons in human obesity disorders and how the activity of these cells is regulated remained poorly studied. We found that *Rai1* deficiency, linked to SMS, regulates BDNF expression and maintains intrinsic excitability in PVH[BDNF] neurons.

Consistent with this mechanism, we found that removing RAI1 from BDNF-producing neurons or PVH[BDNF] neurons results in imbalanced energy homeostasis and obesity. A small molecule drug LM22A-4 led to increased AKT phosphorylation and partially rescued locomotor activity, insulin intolerance, and HDL levels, while delaying the onset of obesity in SMS mice. These findings highlight a potential role for hypothalamic BDNF-producing neurons, specifically PVH[BDNF] neurons, in SMS pathogenesis and highlight BDNF signalling as a potentially druggable pathway to improve energy homeostasis defects and repetitive behaviours commonly observed in SMS patients (*Burns et al., 2010*; *Moss et al., 2009*).

The involvement of different PVH subtypes in human obesity is only beginning to be elucidated. Dysfunction of oxytocin PVH neurons is linked to the Prader–Willi syndrome (*Lee et al., 2012*), another genetic disorder associated with obesity. Since *Rai1* expression is enriched in PVH[BDNF] neurons but not PVH[Oxytocin] neurons, our data suggest that distinct pathophysiological mechanisms underlie SMS and Prader–Willi syndrome.

The conditional knockout analysis allowed us to precisely map the cell-specific origin of neuropathological phenotypes, such as SMS-like symptoms in mice. Our previous work found that deleting *Rai1* from cortical excitatory neurons using an *Emx1[Cre]* allele (*Gorski et al., 2002*), from the GABAergic neurons using a *Gad2[Cre]* allele (*Taniguchi et al., 2011*), or from astrocytes using a *Gfap[Cre]* allele (*Garcia et al., 2004*) did not induce SMS-like obesity (*Huang et al., 2016*). In contrast, obesity could be induced by ablating *Rai1* from Sim1-lineage neurons and, to a lesser extent, SF1-lineage neurons (*Huang et al., 2016*). This analysis suggests that Rai1 expression in PVH is more important than in the ventromedial nucleus of the hypothalamus (VMH) for body weight regulation. Interestingly, although BDNF is required in the VMH and DMH to regulate body weight (*Unger et al., 2007*), embryonic deletion of *Bdnf* from the SF1-lineage populations including the VMH did not result in obesity (*Kamitakahara et al., 2016*; *Yang et al., 2016*). By contrast, deleting *Bdnf* from Sim1-lineage neurons including the PVH resulted in a more severe obesity phenotype (*An et al., 2015*). This is consistent with the idea that BDNF expression in PVH is critical for energy homeostasis. We found that *Rai1* expression in BDNF-producing cells regulates energy homeostasis in a sexually dimorphic manner. Specifically, female cKO mice showed normal food intake, decreased energy expenditure, and reduced locomotor activity. In contrast, male cKO mice showed increased food intake with no changes in locomotor activity. Intriguingly, energy expenditure was increased in male cKO mice, likely due to the higher demand to maintain normal activity levels in obese mice. In both sexes, cKO mice

showed an increased fat mass and adiposity with no change in lean mass, highlighting an essential role for Rai1 in regulating fat deposition. Our data also indicate that BDNF-producing cells are only partially responsible for obesity in SMS: male and female cKO mice become overweight by 15 wk, in contrast to SMS mice which become obese by 9 wk of age, suggesting that non-BDNF neurons contribute to SMS phenotypes.

Notably, mice lacking one copy of *Rai1* in the BDNF-producing cells do not exhibit obesity, whereas SMS patients and SMS mice show pronounced obesity (**Burns et al., 2010**; **Huang et al., 2016**; **Smith et al., 2005**). This indicates that although reduced *Bdnf* expression and BDNF-producing neurons contribute to regulating body weight, additional molecular changes and other hypothalamic populations also play important roles in regulating body weight homeostasis in SMS. Our RPPA data suggest that mTOR signalling is also misregulated in addition to the reduced activation of the neurotrophin downstream cascades. Hypothalamic mTORC1 is crucial to regulate glucose release from the liver, peripheral lipid metabolism, and insulin sensitivity (**Burke et al., 2017**; **Caron et al., 2016**; **Smith et al., 2015**), while mTORC2 regulates glucose tolerance and fat mass (**Kocalis et al., 2014**). How the impaired mTOR signalling contributes to energy homeostasis defects in SMS and the therapeutic potential of targeting this pathway to treat SMS-related obesity remains unclear and warrants future investigation.

What additional RAI1-dependent hypothalamic cell types residing in brain regions other than PVH regulate obesity in SMS? Other important cell types such as TRKB neurons within the PVH (**An et al., 2020**) and several RAI1-expressing hypothalamic nuclei including the arcuate nucleus, VMH, and lateral hypothalamus all play important roles in regulating energy homeostasis. POMC- and AGRP-expressing neurons within the arcuate nucleus are known to regulate food intake and glucose and insulin homeostasis (**Quarta et al., 2021**; **Vohra et al., 2022**). Therefore, *Rai1* function in these neurons could contribute to obesity in SMS, a topic that awaits future investigation.

We found the metabolic phenotypes induced by deleting *Rai1* from BDNF-producing cells were specifically due to *Rai1* deletion in PVH[BDNF] neurons. Therefore, our study focused on PVH[BDNF] neurons. Electrophysiological experiments showed that most wild-type PVH[BDNF] neurons display high-frequency spontaneous AP firing (1–10 Hz) at the resting membrane potential. Most PVH[BDNF] neurons fire sharp APs held at a hyperpolarized potential and show repetitive overshoot during the current ramp test. Loss of *Rai1* significantly reduced the intrinsic neuronal excitability of PVH[BDNF] neurons. Because GABAergic *Rai1* loss did not induce obesity (**Huang et al., 2016**) and most BDNF-producing neurons are glutamatergic (**Canals et al., 2001**). The ionic changes in PVH[BDNF] neurons that precisely explain these excitability changes await future investigations. Different groups of PVH neurons show phasic bursting or slowly inactivating delayed rectifier potassium conductance, which are properties mediated by different potassium channels (**Lee et al., 2012**). Given that our previous RNA-seq experiment found that hypothalamic Rai1 loss induces misexpression of potassium channel subunits *Kcnc2* and *Kcnj11* (**Huang et al., 2016**), RAI1-dependent potassium channels are likely involved in altered neuronal activity. Furthermore, we found fewer low threshold depolarization events in cKO PVH[BDNF] neurons during the hyperpolarized voltage step, suggesting a potential involvement of HCN channels.

Our previous work found that *Rai1* loss in the dentate gyrus resulted in neuronal hyperexcitability (**Chang et al., 2022**). Here we demonstrated that *Rai1* loss in the hypothalamus resulted in neuronal hypoactivity, suggesting that RAI1 regulates neuronal excitability in a cell type-specific manner. Reduced PVH[BDNF] neuronal activity upon *Rai1* loss could potentially explain obesity, consistent with a previous report that found chemogenetic activation of PVH[BDNF] neurons promotes negative energy balance by decreasing feeding and increasing energy expenditure (**Wu and Xu, 2022**).

It is plausible that RAI1 regulates the expression of genes encoding inward rectifier $K^+$ channels, which regulate neuronal activity and potentially energy homeostasis. For instance, KIR6 (a family of ATP-sensitive potassium channels, $K_{ATP}$) is widely expressed in the hypothalamus. Deleting the hypothalamic KIR6.2 subunit impairs $K_{ATP}$ channel function and glucose tolerance (**Miki et al., 2001**; **Parton et al., 2007**). Moreover, reduced expression of hypothalamic GIRK4 (encoding an inwardly rectifying potassium channel) causes obesity (**Perry et al., 2008**). GABAergic neurotransmission from arcuate AGRP-expressing neurons to the PVH neurons is important to increase appetite by favouring hyperphagia (**Atasoy et al., 2012**). Disrupting the composition of these ion channels could contribute to reduced PVH[BDNF] neuronal firing, which awaits further investigations.

Female mice with a conditional knockout of *Rai1* from BDNF-producing neurons do not display a noteworthy difference in food intake. Conversely, their male counterparts exhibit a significant increase in food intake. Although SMS individuals of both genders tend to overeat, male patients who are obese show significantly higher food consumption than their female counterparts (*Gandhi et al., 2022*). This observation raises the possibility that *Rai1* regulates eating behaviours through multiple cell types in the hypothalamus and that a male-specific involvement of BDNF-producing neurons in regulating food intake potentially provides a neurobiological basis for the observed pattern in SMS patients (*Gandhi et al., 2022*). In future work, it would be interesting to dissect the sex-specific role of RAI1 in regulating the functional development of PVH[BDNF] neurons. In addition to a role for RAI1 in regulating neuronal activity, RAI1 may regulate other aspects of PVH[BDNF] neuronal function, including projections to other brain regions to regulate energy homeostasis. In this regard, it would be interesting to dissect how RAI1 controls the circuit connectivity of PVH[BDNF] neurons and how *Rai1* loss in PVH[BDNF] neurons affects nearby PVH[TRKB] neurons that receive BDNF signalling and are linked to body weight regulation (*An et al., 2020*).

At the molecular level, our RPPA data show that *Rai1* depletion disrupted multiple intracellular signalling pathways and resulted in the hypoactivation of specific phospho-signalling proteins. Multiple early studies support the notion that Rai1 directly promotes *Bdnf* expression. First, hypothalamic *Bdnf* mRNA levels were reduced in SMS mice (*Burns et al., 2010*; *Javed et al., 2021*) or mice with *Rai1* conditional deletions (*Huang et al., 2016*). Second, the RAI1 protein directly binds to a promoter region of *Bdnf* (*Huang et al., 2016*). Finally, when *Rai1* expression was induced using CRISPR activation, *Bdnf* mRNA levels were significantly elevated (*Chang et al., 2023*).

To lend further support, here we show that SMS mice have reduced hypothalamic BDNF protein levels, as well as reduced activation of a TRKB downstream target AKT. Reduced AKT activity is frequently associated with insulin resistance and obesity in mice (*Shao et al., 2000*) and children (*Su et al., 2021*). We also found that obesity, increased HDL levels, and insulin insensitivity in SMS mice could be partially alleviated using a small molecule drug (LM22A-4) that promotes AKT phosphorylation. LM22A-4 penetrates the blood–brain barrier reasonably well and has been shown to ameliorate symptoms of other neurological disorders associated with BDNF dysfunction including Rett syndrome, Huntington's disease, Dravet syndrome, and refractory neonatal seizures (*Canals et al., 2004*; *Chang et al., 2006*; *Gu et al., 2022*; *Kron et al., 2012*; *Ogier et al., 2007*). We recognize that while several in vivo studies have demonstrated the potential of LM22A-4 in targeting neurotrophin downstream signalling (*Kron et al., 2014*; *Li et al., 2017*), an in vitro analysis failed to demonstrate the ability of LM22A-4 to activate TRKB directly (*Boltaev et al., 2017*). Therefore, the precise mechanism by which LM22A-4 enhances AKT cascades in the mammalian brain remains unclear and awaits further investigations. In the hypothalamus of SMS mice, LM22A-4 could indirectly engage neurotrophin downstream PI3K-AKT pathway through the G protein-coupled receptor-dependent transactivation of the TRKB receptor (*Domeniconi and Chao, 2010*) or other unknown mechanisms. Moreover, while LM22A-4 may have potential side effects, we found that wild-type mice treated with LM22A-4 did not show a further decrease in body weight, suggesting limited side effects regarding body weight regulation.

The partial rescue of SMS phenotypes that we observed contrasts with our previous findings that found (1) AAV-mediated BDNF overexpression in PVH at 3 wk of age and (2) genetically overexpressing BDNF from early development could fully prevent the development of obesity in SMS mice (*Javed et al., 2021*). Therefore, further increasing TRKB signalling activation and earlier intervention might improve therapeutic outcomes. Alternatively, our RPPA data showed that in addition to disruption of neurotrophin signalling, multiple pathways including mTOR were downregulated in SMS. Increasing hypothalamic mTOR activity has been shown to decrease food intake and body weight (*Cota et al., 2006*). Therefore, it is possible that enhancing both mTOR and TRKB signalling could achieve a more robust rescue of obesity in SMS mice. Nevertheless, these data support the concept that drugging TRKB signalling may improve stereotypical repetitive behaviour and partially alleviate metabolic functions in a mouse model of SMS. Delayed symptom onset could provide a window of opportunity for other interventions to further modify disease progression, an area requiring further investigation.

This study has several limitations. First, whether the PVH[BDNF] neurons depend on Rai1 to regulate adaptive thermogenesis awaits future analysis. It is known that a subset of posterior PVH[BDNF] neurons project to the spinal cord and regulate thermogenesis (*An et al., 2015*). However, we found the

weights of brown adipocytes were similar in control and cKO mice. Moreover, SMS mice have normal respiratory exchange rates and there are currently no reports of defective adaptive thermogenesis in SMS patients. This suggests that adaptive thermogenesis is not impacted in SMS. Another limitation is that we did not test different doses, durations, and time points for LM22A-4 treatment. The precise mechanisms by which LM22A-4 improves body weight, metabolic function, and repetitive behaviours remain unknown. Nevertheless, our work identifies RAI1 as a novel regulator of the neuronal excitability of PVH[BDNF] neurons. This regulation is critical for regulating food intake and energy expenditure. Our evidence suggests that BDNF-producing neurons are involved in SMS pathogenesis and that enhancing BDNF-TRKB signalling should be explored as a future therapeutic avenue for obesity and metabolic conditions associated with reduced neurotrophin signalling (*Xu and Xie, 2016*).

# Materials and methods

## Animal management

The animal protocol (MUHC-8127) was approved by the animal care committee of the Montreal General Hospital and was performed according to the guidelines of the Canadian Council on Animal Care. Mice were group-housed in 12 hr light/12 hr dark cycles with an ad libitum supply of regular chow and water. The exception was in the fasting studies when the animals were food-deprived for 5 hr and in CALMS and food intake studies when animals were single-housed in a 12 hr light/12 hr dark cycle with an ad libitum supply of regular chow and water. All mice were maintained in a C57Bl/6J background. *Bdnf*[Cre/Cre] (RRID:IMSR_JAX:030189), *Rosa26*[Ai9/Ai9] (RRID:IMSR_JAX:007909), and *Actb*[Cre/Cre] (RRID:IMSR_JAX:019099) mice were purchased from the Jackson laboratory. The *Rai1*[fl/fl] (RRID:IMSR_JAX:029103) strain was previously generated and characterized by us (*Huang et al., 2016*). SMS mice (*Rai1*[+/-]: whole-body germline *Rai1* heterozygous knockout that genetically mimics SMS) were generated by crossing germline *Actb*[Cre/Cre] mice with *Rai1*[fl/+]. To generate conditional knockout mice, *Bdnf*[Cre/+]; *Rai1*[fl/+] mice were crossed with *Rai1*[fl/+] to create control and experimental groups from the same littermate. Seven mice with malocclusion or lesions were excluded from the study: four with malocclusion (two females and two males) and three with lesions (three females).

## Metabolic profiling

Metabolic profiling was conducted in three experimental conditions: male and female conditional knockout mice, female virus-injected *Bdnf*[Cre/+] mice, and female drug- or saline-injected SMS and control mice. Metabolic profiling included body weight and body composition analysis, food intake measurement, indirect calorimetry for locomotion activity, energy expenditure (EE), respiratory exchange rate (RER), measurement of serum lipids and leptin levels, adipose tissue histology analysis, and glucose and insulin tolerance tests. All experiments were performed as previously described (*Javed et al., 2021*). A brief description is detailed below.

### Body weight and body composition analysis

The body weight of animals was measured weekly from week 3 (weaning age) onwards. After the 25th week, mice were subjected to body composition analysis. Body composition, including total fat and lean mass, was assessed using a nuclear echo-MRI whole-body analyzer. Towards the end of the experiments, animals were euthanized to collect brown adipose, visceral, subcutaneous (inguinal) and ependymal white adipose tissues, and weighed using an analytical scale (Sartorius).

### Food intake measurement

Mice were single-housed for food intake measurement. Food intake was measured over a 1-week period during which food was weighed prior to placement in the cage with minimal bedding. Each morning, the remaining food inside the cage was weighed. The amount of total food consumed was averaged over 7 d.

### Indirect calorimetry

A Comprehensive Lab Animal Monitoring System (CLAM) (Columbus instruments) was used to measure the RER and EE (normalized over lean mass). Animals were singled-housed in the CLAMS

apparatus at 21°C (70°F) in a light–dark cycle matching their housing conditions for 24 hr (acclimation), followed by 48 hr of measurement.

## Serum lipid level determination

A fluorescence-based assay kit (Abcam ab65390) was used to measure serum lipid levels. The manufacturer's instructions were followed except that we used half-area black 96-well microplates. Fluorescence measurements at 535/587 nm (Ex/Em) were recorded with the Ensight instrument from PerkinElmer. A new standard curve ranging from 0 to 10 μg/ml was produced for each microplate to accurately determine lipid levels in serum samples. Standards and samples were loaded and analysed twice.

## Histology of adipose tissue

Ependymal white adipose tissues (WAT) were excised immediately after the animals were euthanized and fixed with 10% formalin in PBS. The fixed tissues were dehydrated, embedded in paraffin, and sectioned. Two 5-μm-thick sections were taken from each animal and stained with haematoxylin and eosin. Images of sections were then captured using an Aperio ScanScope CS slide scanner at ×20 optical magnification, and the resulting images were saved in jpeg format using the ImageScope software. The images were manually cleaned to remove non-adipose tissues such as lymph nodes. An automated method using the Adiposoft plugin version 1.16 was utilized to determine adipocyte size and number from the jpeg images in Fiji version 2.1.0/1.53c. The Adiposoft settings used excluded cells on edges and set a minimum and maximum cell diameter of 20 and 300 μm, respectively. An experimental value of 2.49 μm per pixel was determined using the Aperio scale bar and used for Adiposoft. To ensure accuracy, two images from each animal were analysed blindly, examining more than 1600 cells per mouse and more than 9100 cells per condition. Adipocyte size frequencies were calculated using the frequency function in Excel.

## Glucose and insulin tolerance tests

For the GTT, experimental mice were food-deprived for 5 hr with ad libitum water supply. A bolus of glucose (1.5 g/kg of lean body weight) was then administered intraperitoneally and glycaemia was measured from tail vein blood samples using the Accu-chek Performa glucometer at T0 (before the injection) as well as at 15, 30, 45, 60, 90, and 120 min (after the injection). Tail vein blood samples were collected via a capillary for insulin assays at T0, 15 and 30 min. For insulin tolerance tests, experimental mice were food-deprived for 5 hr with ad libitum access to water. A bolus of insulin (1 U/kg of lean body weight) was administered via an IP injection and glycaemia was measured from blood sampled at the tail vein using an Accu-chek Performa glucometer at T0 (before the injection), as well as at 15, 30, 45, 60, 90, and 120 min (after the injection). Tail vein blood samples were collected via a capillary for insulin assays at T0.

## Stereotaxic injections

*Bdnf*$^{Cre/+}$ mice were anaesthetized using isoflurane, and the ears were barred onto a stereotaxic device (David Kopf Instruments). AAVs (volume = 250 nl) were bilaterally injected into the PVH using the following coordinates relative to the Bregma: –0.6 mm anteroposterior (AP), ±0.35 mm mediolateral (ML) and –5.52 mm dorsoventral (DV). A description of the AAV constructs and titre used is detailed below:

> AAV9-CMV7-DIO-saCas9 (viral title: >1 × 10$^{13}$), Vector Biolab
> AAV9-CMV7-sgRai1-GFP (viral title: >1 × 10$^9$), Applied Biological Materials Inc. Three sgRNAs were pooled to target Rai1 (sgRNA1: TTCCTCGCCAGAGTAGCGCC; sgRNA2: CCCAGCCT CATGATAGGCCG; sgRNA3: CCCAGCCTCATGATAGGCCG).

## Immunostaining of mouse brain tissues

Immunostaining experiments were performed as previously described (*Huang et al., 2012*). Briefly, mice were anaesthetized and perfused with 1× PBS and 4% formaldehyde, and the brains were dissected and fixed in 4% formaldehyde overnight. Fixed brains were transferred to 30% sucrose solution and subsequently embedded in the OCT solution. The brains were then sectioned (40 μm),

washed in PBS, permeabilized with 0.5% Triton X-100, and incubated with 10% normal donkey serum in PBS for 2 hr. The sections were then incubated overnight with primary antibodies (Rai1: 1:500 dilution, RRID:AB_2921229 [*Huang et al., 2016*]; BDNF: 1:500, RRID:AB_10862052, Abcam ab108319; p-TRKB: 1:500 dilution, RRID:AB_2721199, Millipore ABN1381; oxytocin: 1:1000 dilution, RRID:AB_11212999, Millipore MAB5296) at 4°C on a shaker. The following day, sections were washed with 1× PBS twice for 10 min and incubated with a secondary antibody (1:1000 dilution) for 3 hr at room temperature. Sections were washed with 1× PBS once before mounting on glass slides with the DAPI-mounting (DAPI-Fluoromount-G, SouthernBiotech) and then imaged using a confocal micro-scope (Olympus FV-1000 confocal laser scanning microscope) with ×20 or ×40 objectives (NA 0.85 and 1.3, respectively). Z-stacks were taken with a step size of 1.0 µm with a 1024 × 1024 resolution. Images were acquired as a stack with at least 13 optical sections. Image analysis was performed using the Fiji software.

## Real-time quantitative reverse transcription PCR (qRT-PCR)

qRT-PCR was performed using mRNAs isolated from hypothalamic tissue samples. After the isolation of total RNA, mRNAs were reverse-transcribed with the SuperScript III First-Strand Synthesis System (Thermo Fisher). Quantitative PCR reactions were conducted using SsoFast EvaGreen Supermix on a Bio-Rad qPCR system.

## Enzyme-linked immunosorbent assay (ELISA)

ELISA was performed using hypothalamic tissues from 7-week-old SMS mice to assess the mature BDNF protein levels using a commercially available BDNF ELISA kit (Mature BDNF Rapid ELISA Kit: human, mouse, rat).

## Western blotting analysis

Mice were quickly decapitated, brains were rapidly removed, and hypothalamic tissues were collected, weighed, and snap-frozen with liquid nitrogen. Hypothalamic tissues were lysed using a tissue lysis buffer containing RIPA buffer (5 M NaCl, 1 M PH 8.0 Tris HCl, 1% NP.40, 10% $C_{24}H_{39}NaO_4$, and 20% SDS) and the Halt Protease and Phosphatase Inhibitor Cocktail (Thermo Fisher). Lysates were first soni-cated on ice for 15 s and centrifuged at 10,000 RPM at 4°C for 10 min. The supernatant was collected in a separate Eppendorf tube, and a BCA assay was performed to quantify protein levels. Proteins were then separated using SDS-PAGE gel and transferred onto nitrocellulose membranes. To deter-mine the phosphorylation levels of AKT, membranes were cut and blocked in 5% milk in tris-buffered saline (TBS) for 1 hr and then incubated overnight at 4°C with anti-p-AKT (Ser473) antibody (1:1000 dilution, RRID:AB_2315049, Cell Signaling Technology 4060T) and anti-H3 antibody (internal control: 1:1000 dilution, RRID:AB_302613, Abcam ab1791). The next day, membranes were washed with TBST three times for 5 min before incubating with an HRP-conjugated secondary antibody (1:10,000 dilu-tion, Thermo Fisher 200-403-095S). Membranes were later stripped by incubating in 0.5 M NaOH for 10 min and then re-probed for total Akt levels by incubating with anti-Akt (C67E7) antibody (1:1000 dilution, Cell Signaling Technology 4691T). The density of each immunoreactive phospho-protein band was expressed as a fraction of the band for the total AKT protein in the same lane.

## Reverse-phase protein array (RPPA)

An RPPA was used for precise high-throughput quantification of total and phospho-protein levels in different signalling pathways. Protein lysates from hypothalamic tissues were extracted and trans-ferred to 384-well microarray plates to print individual slides. The slides were then labelled with vali-dated antibodies recognizing total protein and specific phosphorylation sites. SYPRO Ruby staining was used to measure total protein levels. The data obtained from RPPA were normalized for individual samples. Each antibody's raw signal intensity (SI) was determined by subtracting the background from the antibody-specific signals during image analysis. The normalized antibody SI was expressed using the formula $N = A - C/T \times M$, where N is the normalized SI, A is the raw antibody SI, C is the negative control SI, T is the SI of total protein, and M is the median SI of total protein from a spot within the same sample group. Antibodies with SI less than 200 were filtered out, and quality control (QC) scores were established to discard poor-quality reactions. The coefficient of variation (CV) score was determined for each antibody and defined as the percentage of samples on a slide with a CV <

20%. Antibodies with a median technical triplicate CV >20% were also excluded from further analysis. t-tests were performed for each antibody to identify differentially expressed proteins in the SMS compared to control groups. PCA plots were generated for all samples, annotated with their experimental group, and hierarchical clustering was performed using a Python analysis tool. The vertical axis of the dendrogram represents the dissimilarity (measured as distance) between protein expressions, and the horizontal axis represents the individual test samples. The colour code (ranging from red to yellow to green) specifies the expression levels of different proteins, where red indicates low expression, yellow indicates intermediate expression, and green indicates high expression. Enrichment analysis on the differentially expressed proteins was performed using an online data resource string: version 11.5.

## Electrophysiology of acute brain slices

Mouse brains were removed and placed in ice-cold carbonated slicing artificial cerebrospinal fluid (aCSF), which contained (in mM) 124 NaCl, 3 KCl, 1.4 $NaH_2PO_4$, 26 $NaHCO_3$,1.3 Mg SO4, 10 D-glucose, 0.5 $CaCl_2$, and 3.5 $MgCl_2$. Coronal sections (300 μm) were cut on a vibratome. Slices were allowed to recover at 32°C for 40 min and then at 30°C for 30 min to 6 hr in the recording medium, carbonated aCSF (124 NaCl, 3 KCl, 1.4 $NaH_2PO_4$, 26 $NaHCO_3$,1.3 Mg SO4, 10 D-glucose, 2.4 $CaCl_2$, in mM, ~300 mOsm). Signals were recorded with a 5× gain, low-pass filtered at 2 kHz, digitized at 10 kHz (Molecular Devices Multiclamp 700B) and analysed with pClamp 11 (Molecular Devices). Whole-cell recordings were made using 3–5 MΩ pipettes filled with an internal solution that contained (in mM) 131 potassium gluconate, 8 NaCl, 20 KCl, 2 EGTA, 10 HEPES, 2 MgATP, and 0.3 Na3GTP (pH 7.2, with KOH, 300–310 mOsm with sucrose). For miniature current recording, the internal solution contained (in mM) 130 gluconate acid, 8 CsCl, 1 NaCl, 2 EGTA, 0.8 CsOH, 10 HEPES, 2 MgATP, and 0.3 Na3GTP (pH 7.2, with CsOH, 300–310 mOsm adjusted with sucrose). mEPSCs were recorded in the presence of 1 μM TTX at a holding potential of –70 mV. mIPSCs were recorded at a holding potential of +10 mV. The mIPSCs were analysed from events during a 2 min continuous recording that contained >15 events/neuron. Additional calculations, statistical analysis, and plotting were performed in Microsoft Excel or Prism (GraphPad).

## Spectrometric analysis of LM22A-4 levels in the brain

We assessed the efficacy of LM22A-4, a small molecule pharmacological agent, in 8-week-old SMS and control mice. LM22A-4, chemical name N1, N3, N5tris (2hydroxyethyl) 1, 3, 5 benzenetricarboxamide (molecular formula: $C_{15}H_{21}N_3O_6$; formula weight: 339.3), was manufactured by Cayman Chemical with purity specification >98%. A stock solution was made by first dissolving LM22A-4 in DMSO (solubility = 30 mg/ml) and then diluting it with 0.9% isotonic sterilized saline solution (Sigma). To maximize brain concentrations, LM22A-4 was administered both intranasally (5 mg/kg) and intraperitoneally (50 mg/kg), as previously reported (*Simmons et al., 2013*). The bioavailability of this drug in the hypothalamus using these doses and routes of administration was assessed with the reverse-phase LC triple-quadrupole tandem mass spectroscopic detection (LC-MS/MS) by Absorption Systems. Mice were euthanized, and hypothalamic tissue was collected at either 1 hr or 3 hr post-injection (n = 4/ timepoint). Known amounts of LM22A-4 were added to the blank sample to generate a standard curve to verify test accuracy.

## LM22A-4 treatment

Mice were divided into four groups: saline-injected control, saline-injected SMS, LM22A-4-injected SMS, and LM22A-4-injected control mice. Mice were injected daily, intranasally (5 mg/kg) and intraperitoneally (50 mg/kg), for 7 wk. Body weight was measured between 8 and 14 wk of age. Food intake was measured from 14 to 15 wk of age. After 15 wk of age, mice were euthanized to collect brown adipose, visceral, subcutaneous (inguinal), and ependymal white adipose tissues and weighed using an analytical scale (Sartorius). Hypothalamic brain tissues were collected for western blot analysis. Metabolic profiling was performed on a separate cohort of mice injected for 2 wk prior to the start of experiments as well as at specific weeks during the experiment.

## Vertical rearing

To measure vertical repetitive rearing, mice were taken into the testing room and allowed to habituate for 1 hr prior to testing. During the test, mice were placed in the centre of a 45 cm (W) × 45 cm (D) × 40 cm (H) square arena, and their movement was recorded for 10 min.

## Statistical analysis

The sample sizes used were based on our previous studies (*Huang et al., 2016*; *Javed et al., 2021*). The data were subjected to statistical analysis for significance by using GraphPad Prism 0.9. Significance was defined as having a p-value<0.05. The significance levels were as follows: *p<0.05, **p<0.01, ***p<0.0001, **** p<0.0001. The statistical tests used for each analysis (e.g. Student's *t*-test, [ANOVA]) are indicated in the text and legends.

## Acknowledgements

SJ was supported by a Fonds de Recherche du Québec-Santé (FRQS) doctoral award. This work was supported by the Rare Diseases: Models & Mechanisms Network and the Smith-Magenis syndrome Research Foundation.

## Additional information

### Funding

| Funder | Grant reference number | Author |
|---|---|---|
| Fonds de Recherche du Québec - Santé | | Sehrish Javed |
| Rare Disease: Models & Mechanisms Network | | Wei-Hsiang Huang |
| Smith-Magenis Syndrome Research Foundation | | Wei-Hsiang Huang |

The funders had no role in study design, data collection and interpretation, or the decision to submit the work for publication.

### Author contributions

Sehrish Javed, Conceptualization, Data curation, Formal analysis, Investigation, Visualization, Writing – original draft, Writing – review and editing; Ya-Ting Chang, Data curation, Formal analysis, Investigation, Visualization; Yoobin Cho, Yu-Ju Lee, Hao-Cheng Chang, Minza Haque, Yu Cheng Lin, Formal analysis, Investigation; Wei-Hsiang Huang, Conceptualization, Resources, Supervision, Funding acquisition, Writing – original draft, Project administration, Writing – review and editing

### Author ORCIDs

Sehrish Javed  http://orcid.org/0009-0007-8758-8124
Wei-Hsiang Huang  http://orcid.org/0000-0002-8411-9433

### Ethics

The animal protocol (MUHC-8127) was approved by the animal care committee of the Montreal General Hospital and was performed according to the guidelines of the Canadian Council on Animal Care.

Reviewer #1 (Public Review): https://doi.org/10.7554/eLife.90333.3.sa1
Reviewer #2 (Public Review): https://doi.org/10.7554/eLife.90333.3.sa2
Reviewer #3 (Public Review): https://doi.org/10.7554/eLife.90333.3.sa3
Author Response https://doi.org/10.7554/eLife.90333.3.sa4

## Additional files

### Supplementary files
• MDAR checklist

### Data availability
All data generated or analyzed during this study are included in the manuscript and the figure supplement.

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
