## [Editor Report · eLife assessment]

This **valuable** study informs whether diminishing BDNF expression or alterations in the activity of BDNF-containing neurons in the paraventricular nucleus of the hypothalamus contributes to metabolic alterations in individuals with reduced RAI1 function, including those afflicted with Smith–Magenis syndrome (SMS). The evidence supporting the conclusions is **compelling** in that RAI1 deficits in BDNF-containing neurons partly contribute, with prominent effects on glycaemic control and modest effects on feeding and body weight regulation. This study would be of interest to neuroscientists and medical biologists working on metabolic disorders such as obesity and diabetes, as the findings in this study further link SMS-associated obesity with reduced Bdnf gene expression in the PVH and shed light on the role of the Rai1 gene in the PVH Bdnf neurons and offer a basis for future therapeutic strategies for managing obesity in SMS.

---

## [Referee Report · Reviewer #1 (Public Review)]

**Reviewer #1 (Public Review):

**

Summary:

Rai1 encodes the transcription factor retinoic acid-induced 1 (RAI1), which regulates expression of factors involved in neuronal development and synaptic transmission. Rai1 haploinsufficiency leads to the monogenic disorder Smith-Magenis syndrome (SMS), which is associated with excessive feeding, obesity and intellectual disability. Consistent with findings in human subjects, Rai1+/- mice and mice with conditional deletion of Rai1 in Sim+ neurons, which are abundant in the paraventricular nucleus (PVN), exhibit hyperphagia, obesity and increased adiposity. Furthermore, RAI1-deficient mice exhibit reduced expression of brain-derived neurotrophic factor (BDNF), a satiety factor essential for the central control of energy balance. Notably, overexpression of BDNF in PVN of RAI1-deficient mice mitigated their obesity, implicating this neurotrophin in the metabolic dysfunction these animals exhibit. In this follow up study, Javed et al. interrogated the necessity of RAI1 in BDNF+ neurons promoting metabolic health.

Consistent with previous reports, the authors observed reduced BDNF expression in hypothalamus of Rai1+/- mice. Moreover, proteomics analysis indicated impairment in neurotrophin signaling in the mutants. Selective deletion of Rai1 in BDNF+ neurons in the brain during development resulted in increased body weight, fat mass and reduced locomotor activity and energy expenditure without changes in food intake. There was also a robust effect on glycemic control, with mutants exhibiting glucose intolerance. Selective depletion of RAI1 in BDNF+ neurons in PVN in adult mice also resulted in increased body weight, reduced locomotor activity and glucose intolerance without affecting food intake. Blunting RAI1 activity also leads to increases and decreases the inhibitory tone and intrinsic excitability, respectively, of BDNF+ neurons in the PVN.

Overall, the experiments are well designed and multidisciplinary approaches are employed to demonstrate that RAI1 deficits in BDNF+ neurons diminish hypothalamic BDNF signaling and produce metabolic dysfunction. The most significant advance relative to previous reports is the finding from electrophysiological studies showing that blunting RAI1 activity leads to increases and decreases the inhibitory tone and intrinsic excitability, respectively, of BDNF+ neurons in the PVN. Furthermore, that intact RAI1 function is required in BDNF+ neurons for the regulation of glucose homeostasis.

Depleting RAI1 in BDNF+ neurons had a robust effect compromising glycemic control while playing a lesser part driving deficits in energy balance regulation. Accordingly, both global central depletion of Rai1 in BDNF+ neurons during development and deletion of Rai1 in BDNF+ neurons in the adult PVN elicited modest effects on body weight (less than 18% increase) and did not affect food intake. This contrasts with mice with selective Bdnf deletion in the adult PVN, which are hyperphagic and dramatically obese (90% heavier than controls). Therefore, the results suggest that deficits in RAI1 in PVN or the whole brain only moderately affect BDNF actions influencing energy homeostasis and that other signaling cascades and neuronal populations play a more prominent role driving the phenotypes observed in Rai1+/- mice, which are hyperphagic and 95% heavier than controls. The results from the proteomic analysis of hypothalamic tissue of Rai1 mutant mice and controls could be useful in generating alternative hypotheses.

---

## [Referee Report · Reviewer #2 (Public Review)]

Understanding disease conditions often yields valuable insights into the physiological regulation of biological functions, as well as potential therapeutic approaches. In previous investigations, the author's research group identified abnormal expression of brain-derived neurotrophic factor (BDNF) in the hypothalamus of a mouse model exhibiting Smith-Magenis syndrome (SMS), which is caused by heterozygous mutations of the Rai1 gene. Human SMS is associated with distinct facial characteristics, sleep disturbances, behavioral issues, and intellectual disabilities, often accompanied by obesity. Conditional knockout (cKO) of the Bdnf gene from the paraventricular hypothalamus (PVH) in mice led to hyperphagic obesity, while overexpression of the Bdnf gene in the PVH of Rai1 heterozygous mice restored the SMS-like obese phenotype. Based on these preceding findings, the authors of the present study discovered that homozygous Rai1 cKO restricted to Bdnf-expressing cells, or Rai1 gene knockdown solely in Bdnf-positive neurons in the PVH, induced obesity along with intricate alterations in adipose tissue composition, energy expenditure, locomotion, feeding patterns, and glucose tolerance, some of which varied between sexes. Additionally, the authors demonstrated that a brain-penetrating drug capable of activating the AKT cascades, a downstream signaling pathway of BDNF, partially alleviated the SMS-like obesity phenotype in female mice with Rai1 heterozygous mutations. Although the specific (neural) cell type responsible for this signaling remains an open question, the present study unequivocally highlights the importance of Rai1 gene function in PVH Bdnf neurons for the obesity phenotype, providing valuable insights into potential therapeutic strategies for managing obesity associated with SMS.

In the proteomic analysis (Fig. 1), the authors elucidated that multiple phospho-protein signaling pathways, including Akt and mTOR pathways, exhibited significant attenuation in the SMS model mice. Of significance, the manifestation of haploinsufficiency of the Rai1 gene exclusively within the BDNF+ cells demonstrated negligible impact on body weight (Fig. 2-supple 3D), despite observing a reduction in BDNF levels in the heterozygous Rai1 mutant (Fig. 1A). Conversely, the homozygous Rai1 cKO in the BDNF+ cells prominently displayed an obesity phenotype, suggesting substantial dissimilarities in the gene expression profiles between Rai1 heterozygous and homozygous conditions within the BDNF+ cell population. It would be advantageous to precisely identify the responsible differentially expressed genes, possibly including Bdnf itself, in the homozygous cKO model. The observed reduction in the excitability of PVH BDNF+ cells (Fig. 3) is presumably attributed to aberrant gene expression other than Bdnf itself, which may serve as a prospective target for gene expression analysis. Notably, the Rai1 homozygous cKO mice in BDNF+ cells exhibited some sexual dimorphisms in feeding and energy expenditures, as evidenced by Fig. 2 and related figures. Exploring the potential relevance of these sexual differences to human SMS cases and investigating the underlying cellular/molecular mechanisms in the future would provide valuable insights.

The CRISPR-mediated knockdown of the Rai1 gene appears to be highly effective, and the majority of Rai1 cKO effects in Bdnf+ cells are primarily attributed to PVH-Bdnf+ cells based on the similarity of phenotypes observed. With regards to the apparent rescue of the body weight phenotype in Rai1 heterozygous mutants using an AKT pathway activator, the specific biological processes, and neurons responsible for this effect remain unclear. Elucidating these aspects in future studies would be significant when considering potential applications to human SMS cases.

Overall, the present study represents a valuable addition to the authors' series of high-quality molecular genetic investigations into the in vivo functions of the Rai1 gene. This reviewer particularly commends their diligent efforts to enhance our comprehension of SMS and contribute to the future development of more effective therapies for this syndrome.

---

## [Referee Report · Reviewer #3 (Public Review)]

Summary:

Smith-Magenis syndrome (SMS) is associated with obesity and is caused by deletion or mutations in one cope of the Rai1 gene which encodes a transcriptional regulator. Previous studies have shown that Bdnf gene expression is reduced in the hypothalamus of Rai1 heterozygous mice. This manuscript by Javed et al. further links SMS-associated obesity with reduced Bdnf gene expression in the PVH by providing three lines of evidence. First, the authors conducted proteomic analysis of hypothalamic extracts from WT and SMS (Rai1 +/-) mice and showed that several signaling cascades downstream of BDNF (e.g., PI3K-AKT and mTOR) were down regulated in SMS mice. Second, the authors found that deletion of both copies of the Rai1 gene in all BDNF-expressing cells or BDNF-expressing neurons in the PVH led to obesity, although the phenotype is more subtle than that observed in SMS mice. Third, they found that Rai1 deletion reduced excitability of PVH BDNF neurons.

Strengths:

The study provides additional evidence linking BDNF deficiency to hyperphagia and obesity associated with SMS. Furthermore, the study shows that deletion of only one copy of the Rai1 gene in all BDNF-expressing cells did not cause obesity. This result indicates that BDNF deficiency only has a minor contribution to the metabolic symptoms associated with SMS patients who lose one copy of the RAI1 gene. The discovery that Rai1 is important for excitability of PVH BDNF neurons is interesting.

Weaknesses:

The main mechanism underlying SMS-associated obesity remains to be identified. This limitation is discussed in this revised manuscript. The authors also address my previous concerns in this revised manuscript.

---

## [Author Response]

**Reviewer #1 (Public Review):**
Summary:Rai1 encodes the transcription factor retinoic acid-induced 1 (RAI1), which regulates expression of factors involved in neuronal development and synaptic transmission. Rai1 haploinsufficiency leads to the monogenic disorder Smith-Magenis syndrome (SMS), which is associated with excessive feeding, obesity and intellectual disability. Consistent with findings in human subjects, Rai1+/- mice and mice with conditional deletion of Rai1 in Sim+ neurons, which are abundant in the paraventricular nucleus (PVN), exhibit hyperphagia, obesity and increased adiposity. Furthermore, RAI1-deficient mice exhibit reduced expression of brain-derived neurotrophic factor (BDNF), a satiety factor essential for the central control of energy balance. Notably, overexpression of BDNF in PVN of RAI1-deficient mice mitigated their obesity, implicating this neurotrophin in the metabolic dysfunction these animals exhibit. In this follow up study, Javed et al. interrogated the necessity of RAI1 in BDNF+ neurons promoting metabolic health.Consistent with previous reports, the authors observed reduced BDNF expression in the hypothalamus of Rai1+/- mice. Moreover, proteomics analysis indicated impairment in neurotrophin signaling in the mutants. Selective deletion of Rai1 in BDNF+ neurons in the brain during development resulted in increased body weight, fat mass and reduced locomotor activity and energy expenditure without changes in food intake. There was also a robust effect on glycemic control, with mutants exhibiting glucose intolerance. Selective depletion of RAI1 in BDNF+ neurons in PVN in adult mice also resulted in increased body weight, reduced locomotor activity, and glucose intolerance without affecting food intake. Blunting RAI1 activity also leads to increases and decreases in the inhibitory tone and intrinsic excitability, respectively, of BDNF+ neurons in the PVN.Strengths:Overall, the experiments are well designed and multidisciplinary approaches are employed to demonstrate that RAI1 deficits in BDNF+ neurons diminish hypothalamic BDNF signaling and produce metabolic dysfunction. The most significant advance relative to previous reports is the finding from electrophysiological studies showing that blunting RAI1 activity leads to increases and decreases the inhibitory tone and intrinsic excitability, respectively, of BDNF+ neurons in the PVN. Furthermore, that intact RAI1 function is required in BDNF+ neurons for the regulation of glucose homeostasis.Weaknesses:Some of the data need to be reconciled with previous findings by others. For example, the authors report that more than 50% of BDNF+ neurons in PVN also express pTrkB whereas about 20% of pTrkB+ cells contain BDNF, raising the possibility that autocrine mechanisms might be at play. This is in conflict with a previous study by An et al, (2015) showing that these cell populations are largely non-overlapping in the PVN.

We fully agree with this assessment. Given the difficulty of using immunostaining to characterize the expression of membrane proteins in vivo, and the specificity of the pTrkB antibody in different tissues remains unknown, it is difficult to interpret the signals we observed. We have excluded the data because the histological analysis of p-TRKB and BDNF autocrine/paracrine signalling is not a focus of the present study. Future studies using a more advanced genetic method (i.e., Ntrk2CreER/+; Ai9 mouse line as used by An et al., 2015) is more suitable and should be used in the future to investigate the function of Rai1 in the TRKB+ neurons.

Another issue that deserves more in-depth discussion is that diminished BDNF function appears to play a minor part driving deficits in energy balance regulation. Accordingly, both global central depletion of Rai1 in BDNF+ neurons during development and deletion of Rai1 in BDNF+ neurons in the adult PVN elicited modest effects on body weight (less than 18% increase) and did not affect food intake. This contrasts with mice with selective Bdnf deletion in the adult PVN, which are hyperphagic and dramatically obese (90% heavier than controls). Therefore, the results suggest that deficits in RAI1 in PVN or the whole brain only moderately affect BDNF actions influencing energy homeostasis and that other signaling cascades and neuronal populations play a more prominent role driving the phenotypes observed in Rai1+/- mice, which are hyperphagic and 95% heavier than controls. The results from the proteomic analysis of hypothalamic tissue of Rai1 mutant mice and controls could be useful in generating alternative hypotheses. Depleting RAI1 in BDNF+ neurons had a robust effect compromising glycemic control. However, as the approach does not necessarily impact BDNF exclusively, there should be a larger discussion of alternative mechanisms.

We thank the reviewer for these insightful comments. We want to highlight that global deletion of Rai1 from BDNF neurons did induce food intake increase in male mice (Fig 2figure supplement 4K). We have incorporated the following paragraphs into the discussion section.

Lines 364-384: “Notably, mice lacking one copy of Rai1 in the BDNF-producing cells do not exhibit obesity, whereas SMS patients and SMS mice show pronounced obesity (Burns et al., 2010; Huang et al., 2016; Smith et al., 2005). This indicates that although reduced Bdnf expression and BDNF-producing neurons contribute to regulating body weight, additional molecular changes and other hypothalamic populations also play important roles in regulating body weight homeostasis in SMS. Our RPPA data suggest that mTOR signalling is also misregulated in addition to the reduced activation of the neurotrophin downstream cascades. Hypothalamic mTORC1 is crucial to regulate glucose release from the liver, peripheral lipid metabolism, and insulin sensitivity (Burke et al., 2017; Caron et al., 2016; Smith et al., 2015), while mTORC2 regulates glucose tolerance and fat mass (Kocalis et al., 2014). How the impaired mTOR signalling contributes to energy homeostasis defects in SMS and the therapeutic potential of targeting this pathway to treat SMS-related obesity remains unclear and warrants future investigation.

What additional Rai1-dependent hypothalamic cell types residing in brain regions other than PVH regulate obesity in SMS? Other important cell types such as TRKB neurons within the PVH (An et al., 2020) and several RAI1-expressing hypothalamic nuclei including the arcuate nucleus, ventromedial nucleus of the hypothalamus (VMH), and lateral hypothalamus all play important roles in regulating energy homeostasis. POMC- and AGRP-expressing neurons within the arcuate nucleus are known to regulate food intake and glucose and insulin homeostasis (Quarta et al., 2021; Vohra et al., 2022). Therefore, Rai1 function in these neurons could contribute to obesity in SMS, a topic that awaits future investigation.”

**Reviewer #2 (Public Review):**
Understanding disease conditions often yields valuable insights into the physiological regulation of biological functions, as well as potential therapeutic approaches. In previous investigations, the author's research group identified abnormal expression of brain-derived neurotrophic factor (BDNF) in the hypothalamus of a mouse model exhibiting Smith-Magenis syndrome (SMS), which is caused by heterozygous mutations of the Rai1 gene. Human SMS is associated with distinct facial characteristics, sleep disturbances, behavioral issues, and intellectual disabilities, often accompanied by obesity. Conditional knockout (cKO) of the Bdnf gene from the paraventricular hypothalamus (PVH) in mice led to hyperphagic obesity, while overexpression of the Bdnf gene in the PVH of Rai1 heterozygous mice restored the SMS-like obese phenotype. Based on these preceding findings, the authors of the present study discovered that homozygous Rai1 cKO restricted to Bdnf-expressing cells, or Rai1 gene knockdown solely in Bdnf-positive neurons in the PVH, induced obesity along with intricate alterations in adipose tissue composition, energy expenditure, locomotion, feeding patterns, and glucose tolerance, some of which varied between sexes. Additionally, the authors demonstrated that a brain-penetrating drug capable of activating the TrkB pathway, a downstream signaling pathway of BDNF, partially alleviated the SMS-like obesity phenotype in female mice with Rai1 heterozygous mutations. Although the specific (neural) cell type responsible for this TrkB signaling remains an open question, the present study unequivocally highlights the importance of Rai1 gene function in PVH Bdnf neurons for the obesity phenotype, providing valuable insights into potential therapeutic strategies for managing obesity associated with SMS.In the proteomic analysis (Fig. 1), the authors elucidated that multiple phospho-protein signaling pathways, including Akt and mTOR pathways, exhibited significant attenuation in the SMS model mice. Of significance, the manifestation of haploinsufficiency of the Rai1 gene exclusively within the BDNF+ cells demonstrated negligible impact on body weight (Fig. 2supple 3D), despite observing a reduction in BDNF levels in the heterozygous Rai1 mutant (Fig. 1A). Conversely, the homozygous Rai1 cKO in the BDNF+ cells prominently displayed an obesity phenotype, suggesting substantial dissimilarities in the gene expression profiles between Rai1 heterozygous and homozygous conditions within the BDNF+ cell population. It would be advantageous to precisely identify the responsible differentially expressed genes, possibly including Bdnf itself, in the homozygous cKO model. The observed reduction in the excitability of PVH BDNF+ cells (Fig. 3) is presumably attributed to aberrant gene expression other than Bdnf itself, which may serve as a prospective target for gene expression analysis. Notably, the Rai1 homozygous cKO mice in BDNF+ cells exhibited some sexual dimorphisms in feeding and energy expenditures, as evidenced by Fig. 2 and related figures. Exploring the potential relevance of these sexual differences to human SMS cases and investigating the underlying cellular/molecular mechanisms in the future would provide valuable insights.Although the CRISPR-mediated knockdown of the Rai1 gene (Fig. 4) appears to be highly effective, given the broad transduction of AAV serotype 9, it may be helpful to exclude the possibility of other brain regions adjacent to the PVH, such as the DMH or VMH, being affected by this viral procedure. If the PVH-specificity is established, the majority of Rai1 cKO effects in Bdnf+ cells are primarily attributed to PVH-Bdnf+ cells based on the similarity of phenotypes observed. With regards to the apparent rescue of the body weight phenotype in Rai1 heterozygous mutants using a selective TrkB activator, the specific biological processes, and neurons responsible for this effect remain unclear to this reviewer. Elucidating these aspects would be significant when considering potential applications to human SMS cases.

We appreciate the reviewer's insightful comments. We agree that the logical next step would be to identify the profile of the differentially expressed genes in our homozygous conditional knockout model. We have included the following paragraphs in the discussion.

Lines 364-384: “Notably, mice lacking one copy of Rai1 in the BDNF-producing cells do not exhibit obesity, whereas SMS patients and SMS mice show pronounced obesity (Burns et al., 2010; Huang et al., 2016; Smith et al., 2005). This indicates that although reduced Bdnf expression and BDNF-producing neurons contribute to regulating body weight, additional molecular changes and other hypothalamic populations also play important roles in regulating body weight homeostasis in SMS. Our RPPA data suggest that mTOR signalling is also misregulated in addition to the reduced activation of the neurotrophin downstream cascades. Hypothalamic mTORC1 is crucial to regulate glucose release from the liver, peripheral lipid metabolism, and insulin sensitivity (Burke et al., 2017; Caron et al., 2016; Smith et al., 2015), while mTORC2 regulates glucose tolerance and fat mass (Kocalis et al., 2014). How the impaired mTOR signalling contributes to energy homeostasis defects in SMS and the therapeutic potential of targeting this pathway to treat SMS-related obesity remains unclear and warrants future investigation.

What additional Rai1-dependent non-PVH hypothalamic cell types regulate obesity in SMS? Other important cell types such as TRKB neurons within the PVH (An et al., 2020) and several RAI1expressing hypothalamic nuclei including the arcuate nucleus, ventromedial nucleus of the hypothalamus (VMH), and lateral hypothalamus all play important roles in regulating energy homeostasis. POMC- and AGRP-expressing neurons within the arcuate nucleus are known to regulate food intake and glucose and insulin homeostasis (Quarta et al., 2021; Vohra et al., 2022). Therefore, Rai1 function in these neurons could contribute to obesity in SMS, a topic that awaits future investigation.”

Lines 409-418: “It is plausible that RAI1 regulates the expression of genes encoding inward rectifier K+ channels, which regulate neuronal activity and potentially energy homeostasis. For instance, KIR6 (a family of ATP-sensitive potassium channels, KATP) is widely expressed in the hypothalamus. Deleting the hypothalamic KIR6.2 subunit impairs KATP channel function and glucose tolerance (Miki et al., 2001; Parton et al., 2007). Moreover, reduced expression of hypothalamic GIRK4 (encoding an inwardly rectifying potassium channel) causes obesity (Perry et al., 2008). GABAergic neurotransmission from arcuate AGRP-expressing neurons to the PVH neurons is important to increase appetite by favouring hyperphagia (Atasoy et al., 2012). Disrupting the composition of these ion channels could contribute to reduced PVHBDNF neuronal firing, which awaits further investigations.”

Moreover, to facilitate the future exploration of the potential relevance of sexual differences to human SMS cases, we have incorporated the following explanation in the discussion section.

Lines 419-426: “Female mice with a conditional knockout of Rai1 from BDNF-producing neurons do not display a noteworthy difference in food intake. Conversely, their male counterparts exhibit a significant increase in food intake. Although SMS individuals of both genders tend to overeat, male patients who are obese show significantly higher food consumption than their female counterparts (Gandhi et al., 2022). This observation raises the possibility that Rai1 regulates eating behaviours through multiple cell types in the hypothalamus and that a male-specific involvement of BDNF-producing neurons in regulating food intake, potentially provides a neurobiological basis for the observed pattern in SMS patients (Gandhi et al., 2022).”

To exclude the possibility of other brain regions adjacent to the PVH (such as VMH and arcuate nucleus) being affected by our AAV-CRISPR-mediated Rai1 knockout, we have analyzed other hypothalamic regions including VMH and arcuate nucleus from the same slides used to confirm PVH viral expression and we confirmed that the AAV was not expressed in these regions. We have incorporated a representative image (Figure 4 suppl 1F) depicting limiting AAV expression in these nuclei.

Regarding LM22A-4: It is possible that LM22A-4 functions directly through binding to TRKB or indirectly engages TRKB downstream molecules through activating other receptors such as GPCR. LM22A-4 appears to engage neurotrophin downstream PI3KAKT pathway, which was identified by our RPPA analysis to be downregulated in the hypothalamus of Rai1-deficient mice. Reduced AKT activity is associated with insulin resistance and obesity in mice. Restoration of functional activity of AKT by LM22A-4 could be the primary mode of action for this drug in the brain. However, since we observed that this drug only partially rescued the body weight defect, future research exploring more potent TrkB agonists or utilizing a combination therapy that targets both the neurotrophin and mTOR pathways might yield improved responses to the pharmacological interventions. We have included the following paragraph in the discussion:

Lines 451-461: “ We recognize that while several in vivo studies have demonstrated the potential of LM22A-4 in targeting neurotrophin downstream signalling (Kron et al., 2014; Li et al., 2017), an in vitro analysis failed to demonstrate the ability of LM22A-4 to activate TrkB directly (Boltaev et al., 2017). Therefore, the precise mechanism by which LM22A-4 enhances AKT cascades in the mammalian brain remains unclear and awaits further investigations. In the hypothalamus of SMS mice, LM22A-4 could indirectly engage neurotrophin downstream PI3KAKT pathway through the G protein-coupled receptor-dependent transactivation of the TRKB receptor (Domeniconi & Chao, 2010) or other unknown mechanisms. Moreover, while LM22A4 may have potential side effects, we found that wild-type mice treated with LM22A-4 did not show a further decrease in body weight, suggesting limited side effects regarding body weight regulation.”

Overall, the present study represents a valuable addition to the authors' series of high-quality molecular genetic investigations into the in vivo functions of the Rai1 gene. This reviewer particularly commends their diligent efforts to enhance our comprehension of SMS and contribute to the future development of more effective therapies for this syndrome.

We thank the reviewer for finding our study valuable in advancing the understanding of RAI1 function.

**Reviewer #3 (Public Review):**
Summary:Smith-Magenis syndrome (SMS) is associated with obesity and is caused by deletion or mutations in one copy of the Rai1 gene which encodes a transcriptional regulator. Previous studies have shown that Bdnf gene expression is reduced in the hypothalamus of Rai1 heterozygous mice. This manuscript by Javed et al. further links SMS-associated obesity with reduced Bdnf gene expression in the PVH.Strengths:The authors show that deletion of the Rai1 gene in all BDNF-expressing cells or just in the PVH BDNF neurons postnatally caused obesity. Interestingly, mutant mice displayed sexual dimorphism in the cause for the obesity phenotype. Overall, the data are well presented and convincing except the data from LM22A-4.Weaknesses:1. The most serious concern is about data from LM22A-4 administration experiments (Figure 5 and associated supplemental figures). A rigorous study has demonstrated that LM22A-4 does not activate TrkB (Boltaev et al., Science Signaling, 2017), which is consistent with unpublished results from many labs in the neurotrophin field. It is tricky to interpret body weight data from pharmacological studies because compounds always have some side effects, which can reduce body weight non-specifically.

We thank this reviewer for their valuable comments. Indeed, the precise mechanism by which LM22A-4 exerts its effect is not entirely clear and there has been mixed evidence regarding its identity as a TRKB agonist in vitro. We have refrained from stating LM22A-4 as a partial agonist of TRKB, and instead have focused on highlighting the potential of this drug in activating neurotrophin downstream signalling through increasing AKT phosphorylation in vivo. We have modified the title to remove TRKB, and the following changes have been made in the discussion:

Lines 451-461: “ We recognize that while several in vivo studies have demonstrated the potential of LM22A-4 in targeting neurotrophin downstream signalling (Kron et al., 2014; Li et al., 2017), an in vitro analysis failed to demonstrate the ability of LM22A-4 to activate TrkB directly (Boltaev et al., 2017). Therefore, the precise mechanism by which LM22A-4 enhances AKT cascades in the mammalian brain remains unclear and awaits further investigations. In the hypothalamus of SMS mice, LM22A-4 could indirectly engage neurotrophin downstream PI3KAKT pathway through the G protein-coupled receptor-dependent transactivation of the TRKB receptor (Domeniconi & Chao, 2010) or other unknown mechanisms. Moreover, while LM22A4 may have potential side effects, we found that wild-type mice treated with LM22A-4 did not show a further decrease in body weight, suggesting limited side effects regarding body weight regulation.”

1. The resolution of all figures are poor, and thus I could not judge the quality of the micrographs.

We have updated with higher resolution images.

1. Citation of the literature is not precise. The study by An et al. (2015) shows that deletion of the Bdnf gene in the PVH leads to obesity due to increased food intake and reduced energy expenditure (not just hyperphagic obesity; Line 72). Furthermore, the study by Unger et al. (2017) carried out Bdnf deletion in the VMH and DMH using AAV-Cre and did not discuss SF1 neurons at all (Line 354). The two studies by Yang et al. (Mol Endocrinol, 2016) and Kamitakahara et al. (Mol Metab, 2015) did use SF1-Cre to delete the Bdnf gene and did not observe any obesity phenotype.

We thank the reviewer for bringing this to our attention. We have revised the text to ensure accurate representation of the cited publications. The following changes have been made:Lines 348-350: “ Although BDNF is required in the VMH and DMH to regulate body weight(Unger et al., 2007), embryonic deletion of Bdnf from the SF1-lineage populations including theVMH did not result in obesity (Kamitakahara et al., 2016; Yang et al., 2016).”

1. Animal number is not described in many figure legends.

We thank the reviewer for pointing it out. We have revised the manuscript to incorporate the missing animal numbers.

**Reviewer #1 (Recommendations For The Authors):**
Additional points:1. The data provided indicating increased inhibitory tone onto BDNF neurons in PVN of Rai1 mutant mice are not convincing that inhibitory drive is significantly affected.

We have modified the sentences as follows, we have also deleted these conclusions from the abstract and discussion:

Lines 215-220: “We observed a slight rightward shift of the probability of miniature inhibitory postsynaptic current (mIPSC) frequency in cKO PVHBDNF neurons, although the average frequency (Fig 3K) was not significantly different between groups. The probability of mIPSC amplitude also showed a right shift without a significant change (Fig 3L, Figure 3—figure supplement 1D). However, we observes a significant increased area under the curve (Fig 3M).”

1. Fig. 3C - Was outlier analysis performed for these data? One of the data points for the control group looks like an outlier that might be skewing the data.

We performed an outlier analysis and found that indeed one data point was an outlier, after removing this data point, the data remained statistically significant (*p<0.05) and the new manuscript has been updated.

**Reviewer #2 (Recommendations For The Authors):**
1. The manuscript would benefit from improved usage and precise descriptions of statistics. The authors often provided only general statements such as "one or two-way ANOVA" without specifying the exact statistical tests used. It is important to differentiate between one-way and two-way ANOVA, particularly when using the latter, by clearly indicating the within-group effects and interaction effects. The representation of p-values associated with ANOVA using asterisks requires clarification, specifying which statistics indicate ANOVA results and which ones correspond to post hoc analysis. It is advisable to assess the normality of the distribution before employing t-tests or consider non-parametric comparisons such as Wilcoxon's rank sum test if normality assumptions are not met. Additionally, it is essential to specify whether the tests are one-sided or two-sided and whether they are paired or unpaired. In some figure panels, such as Fig. 2H and K, the statistical tests used were not indicated at all.

We have clarified the exact statistical tests in the figure legend for each figure.

1. Rearranging the figures to facilitate a direct comparison of the sexual phenotypes (Fig. 2 and Fig. 2-supple 4) within the same figures would greatly improve reader comprehension.

We have decided to keep the figure arrangement because of the focus on female mice in the main figures.

1. To improve the comprehension of the figures and text, the following points should be addressed:Fig. 1D: The definition of the expression level in the color code is not clear.

Explanation for the color code has been added in the method section.

Lines 652-656: “The vertical axis of the dendrogram represents the dissimilarity (measured as distance) between protein expressions, and the horizontal axis represents the individual test samples. The colour code (ranging from red to yellow to green) specifies the expression levels of different proteins, where red indicates nifies low expression, yellow indicates intermediate expression, and green indicates high expression.”

Fig. 1F: One parenthesis is missing from the figure label.

Fixed

Fig. 2C: It is unclear why there are so many dots for just n = 3 animals. It would be better to specify the conditions or use "animals" as a unit of measurement.

The dots represent percentage cells quantified per sliced from 3 animals. It has been clarified in the figures.

Fig. 2F: There seems to be an unnecessary label "I" in the middle of the panel.

Fixed

It is not completely clear if the data in Fig. 2E-L were all obtained at 26 weeks of age.

To clarify, following line has been added to the method section:

Lines 517-518: “After the 25th week, mice were subjected to body composition analysis.”

In Fig. 2-Supple 1, the legend should read "G-J." Additionally, please provide a definition for the arrowheads.

Line 1086: “yellow arrowheads indicate Ai9 marked BDNF cells co-expressing endogenous BDNF.”

It is not completely clear if the data in Fig. 3 were all obtained from female mice.

It is explained in the legend of Fig 3.

The description of the number of animals seems to be missing in Fig. 4

The description for the number of animals has been added in the figure legend. Line 1004: “(Ctrl group: n = 5, Exp group: n = 5)”

On line 280-281, "Fig 4A." should be corrected to "Fig. 5A."

Corrected.

In Fig. 5C-E, it is uncertain if multiple pairwise comparisons for three groups are statistically appropriate. At the very least, multiple comparisons should be corrected.

We performed two-way ANOVA where mean body weight of age-matched groups were compared with each other (i.e. between control saline-injected and SMS saline-injected,SMS saline-injected and LM22A-4 -saline injected, and Control saline-injected and SMS LM22A-4 injected). We used Šidák’s multiple comparisons test, where statistical significance was indicated with *p<0.05, **p < 0.01, ***p<0.001, ****p < 0.0001. We have clarified this in the figure 5 legends.

The unit of measurement should be standardized across figures, if possible, to facilitate better side-by-side comparisons. For example, most bodyweight figures use "g" (grams), but "mg" (milligrams) is used in Fig. 5.

All measurements are corrected to be consistent (in grams).

It is unclear if nM (not mM) of glucose was actually measured in the glucose tolerance test (Fig. 2L and Fig. 4L).

Fixed.

**Reviewer #3 (Recommendations For The Authors):**
1. The authors can remove the LM22A-4 data without much detrimental effects on the conclusion of the manuscript. Otherwise, the authors have to demonstrate that LM22A-4 activates TrkB, does not have any toxicity, and does not cause aversion.

We thank this reviewer the valuable comments and we acknowledge the valid concern. Indeed, the precise mechanism by which LM22A-4 exert its effects is not clear and there has been mixed opinions regarding its function as TRKB agonist in in-vitro assays. To clarify, we have refrained from stating LM22A-4 as a partial agonist of TRKB, and instead have focused on highlighting the potential of this drug in activating neurotrophin downstream signalling through increased AKT phosphorylation, in-vivo.

We have also modified the title of our article to exclude the word “TRKB Signalling”. The new title is as follows:

“Smith-Magenis syndrome protein RAI1 regulates body weight homeostasis through hypothalamic BDNF-producing neurons and neurotrophin downstream signalling”

1. Line 50: "40% > 95th percentile weight, 40% > 85th percentile weight" should be "40% > 95th percentile weight, 80% > 85th percentile weight".

Corrected.

1. Abbreviations for brain-derived neurotrophic factor: Bdnf for gene and BDNF for protein.

Abbreviations have been corrected throughout the manuscript.

1. Need to specify the animal age when viruses were injected into the PVH to inactivate the Bdnf gene.

Line 235: Virus was injected at 3 weeks of age. It has been specified in the main text.

1. Line 832: "3 technical triplicates" can be simplified as "3 technical repeats" because 3 and triplicates are redundant.

Corrected.

1. Figure 2B: The "O" in cKO is misplaced.

Fixed.

1. Figure 3: The black legends in E and F should include Ctrl.

Fixed in the Figure 3.